# THE QUEST FOR GENERALIZABLE MOTION GENERATION: DATA, MODEL, AND EVALUATION

**Jing Lin**[1*], **Ruisi Wang**[2*], **Junzhe Lu**[3*], **Ziqi Huang**[1], **Guorui Song**[3], **Ailing Zeng**[4],
**Xian Liu**[5], **Chen Wei**[2], **Wanqi Yin**[2], **Qingping Sun**[2], **Zhongang Cai**[2†], **Lei Yang**[2✉], **Ziwei Liu**[1✉]

[1] Nanyang Technological University   [2] SenseTime Research   [3] Tsinghua University
[4] The Chinese University of Hong Kong   [5] NVIDIA Research

jing026@e.ntu.edu.sg   wangruisi1@sensetime.com   lu-jz24@mails.tsinghua.edu.cn
{caizhongang, yanglei}@sensetime.com   ziwei.liu@ntu.edu.sg

## ABSTRACT

Despite recent advances in 3D human motion generation (MoGen) on standard benchmarks, existing text-to-motion models still face a fundamental bottleneck in their *generalization* capability. In contrast, adjacent generative fields, most notably video generation (ViGen), have demonstrated remarkable generalization in modeling human behaviors, highlighting transferable insights that MoGen can leverage. Motivated by this observation, we present a comprehensive framework that systematically transfers knowledge from ViGen to MoGen across three key pillars: data, modeling, and evaluation. First, we introduce **ViMoGen-228K**, a large-scale dataset comprising 228,000 high-quality motion samples that integrates high-fidelity optical MoCap data with semantically annotated motions from web videos and synthesized samples generated by state-of-the-art ViGen models. The dataset includes both text–motion pairs and text–video–motion triplets, substantially expanding semantic diversity. Second, we propose **ViMoGen**, a flow-matching-based diffusion transformer that unifies priors from MoCap data and ViGen models through gated multimodal conditioning. To enhance efficiency, we further develop **ViMoGen-light**, a distilled variant that eliminates video generation dependencies while preserving strong generalization. Finally, we present **MBench**, a hierarchical benchmark designed for fine-grained evaluation across motion quality, prompt fidelity, and generalization ability. Extensive experiments show that our framework significantly outperforms existing approaches in both automatic and human evaluations. The code, data, and benchmark will be made publicly available.

## 1 INTRODUCTION

Breakthroughs in generative models have profoundly transformed human visual creativity, opening new avenues for artistic expression and content production across modalities, including text (Touvron et al., 2023; Guo et al., 2025), image (Rombach et al., 2021; Labs, 2024), and video (Hong et al., 2022; Wang et al., 2025). A critical factor underlying these successes is their strong generalization capability: such models not only excel within their training domains but also extend effectively to novel instructions, thereby empowering human creativity in a broad spectrum of applications. In contrast, 3D human motion generation (MoGen)—specifically in the text-to-motion domain—remains comparatively underdeveloped, particularly in its capacity to generalize to diverse and long-tail instructions. In this work, we propose a holistic approach toward generalizable 3D human motion generation, advancing the field through coordinated innovations in **data**, **modeling**, and **evaluation**. Our effort is guided by a broader vision: to unify and transfer insights from adjacent generative fields, particularly video generation (ViGen), while building motion-specific innovations that lay the groundwork for future exploration toward a general-purpose motion foundation model. While

---

*Equal contribution; †Project Lead; ✉Corresponding Author.

motion generation broadly encompasses audio and music modalities, our study focuses exclusively on text-to-motion synthesis to address the critical semantic generalization gap.

Figure 1: **Overview of our approach toward generalizable 3D human motion generation.** (a) **ViMoGen**: Our model demonstrates superior generalization on challenging prompts including martial arts, dynamic sports, and multi-step behaviors. (b) **MBench**: Comprehensive benchmark evaluating models across dimensions, showing ViMoGen's significant improvements over existing methods. (c) **ViMoGen-228K**: Large-scale dataset with 228,000 motion sequences from diverse sources covering simple indoor to complex outdoor activities.

**First**, we curate **ViMoGen-228K**, a large-scale, diversely sourced, and carefully balanced dataset designed to enable generalizable motion generation. A major bottleneck for motion generation lies in data scarcity: unlike other modalities such as text, image, and video (where massive datasets are readily available online), motion data is orders of magnitude smaller, leading to severely limited semantic coverage. To overcome this, we source heterogeneous data and design a meticulous curation pipeline. ViMoGen-228K ultimately comprises 171.5K text–motion pairs and 56.6K text–video–motion triplets from three complementary sources. (1) Optical motion capture (MoCap) datasets, which, despite physical constraints, provide high-quality motion signals that serve as critical priors for modeling human dynamics. We standardize 30 MoCap datasets, aligning conventions and augmenting them with text annotations. (2) In-the-wild video–derived motions, which significantly broaden semantic coverage. Following prior efforts (Wang et al., 2024c; Liang et al., 2024a; Lin et al., 2023), we construct a rigorous cascaded filtering pipeline that enforces a strict selection scheme: only about ∼1% of motions extracted from a large-scale internal video database are retained, ensuring motion fidelity. (3) Synthetic video–derived motions, generated by a state-of-the-art video generation model (Wan et al., 2025) trained on billions of images and videos, where humans constitute a substantial subset. Unlike scraping Internet videos, leading ViGen models can produce surprisingly realistic, strategically controlled human videos that are easier to capture with MoCap pipelines, when prompted with crafted long-tail instructions to extend behavioral coverage.

**Second**, we introduce **ViMoGen**, a versatile diffusion transformer (DiT)-based model designed for generalizable motion generation. The central idea of ViMoGen is to harness knowledge from proxy generation tasks that have already demonstrated strong human-centric generalization (most notably ViGen models, which excel at producing high-fidelity human motions). Yet, effectively exploiting diverse and heterogeneous knowledge sources remains underexplored: existing frameworks are often inadequate for systematic knowledge transfer, limiting their ability to generalize across contexts. For instance, prior works (Millán et al., 2025; Albaba et al., 2025) rely on costly optimization-based pipelines to extract motion knowledge from ViGen models. In contrast, ViMoGen employs a *gated fusion* and *dual-branch* design that balances complementary contributions from optical MoCap, in-the-wild videos, and synthetic video data: (i) a Text-to-Motion (T2M) branch that conditions generation on text paired with MoCap priors, and (ii) a Motion-to-Motion (M2M) branch that leverages ViGen-derived motion tokens to broaden semantic coverage. This unified architecture enables more efficient and integrated transfer of ViGen priors into the motion diffusion process, combining the semantic richness of video models with the high fidelity of motion-specific synthesis. Furthermore, we develop ViMoGen-light, a lightweight distilled variant that inherits knowledge

directly from ViGen models while bypassing expensive text-to-video inference. As illustrated in Fig. 1, both ViMoGen and ViMoGen-light substantially improve generalization over prior approaches.

**Third**, we present **MBench**, a comprehensive benchmark for accurate and fine-grained evaluation of motion generation algorithms, with a particular emphasis on generalization capability. The field currently lacks a unified evaluation suite that can systematically assess different models, measure various aspects of the generated motions, and provide actionable feedback for researchers. Existing evaluation protocols (Guo et al., 2022a) primarily rely on distribution-based metrics (*e.g.*, FID), which reduce model performance into a single score, sacrificing granularity and often diverging from human judgment (Wang et al., 2024a). Moreover, commonly used test prompts (Guo et al., 2022a; Punnakkal et al., 2021) are dominated by simple, indoor actions, offering limited insight into generalization. To address these gaps, MBench provides a comprehensive and structured assessment framework that it evaluates generated motions along *three* principal axes: motion quality, motion-condition consistency, and generalization ability. In particular, we emphasize the use of a carefully curated open-world vocabulary to rigorously assess generalization beyond conventional action categories. Overall, MBench spans *nine* distinct dimensions, each guided by mindfully designed prompts tailored to its specific characteristics. Importantly, the benchmark undergoes human validation to confirm that the automated components closely align with human judgments and preferences.

We also reveal some valuable insights that we find during our extensive experiments. Our contributions can be summarized as:

- We curate ViMoGen-228K, a large-scale human motion dataset comprising both text-video-motion triplets and text-motion pairs. The dataset features both broad semantic coverage and motion quality.

- We propose ViMoGen, a DiT-based model with innovative gating and masking mechanisms that seamlessly integrate knowledge from multiple modalities while preserving learnet motion priors, along with its efficient distilled variant, ViMoGen-light.

- We build MBench, a comprehensive, fine-grained, and human-aligned benchmark that effectively assesses motion generalization, motion-condition consistency, and motion quality.

## 2 METHOD

We present a comprehensive framework for generalizable text-driven human motion generation, built on a flow-based diffusion transformer architecture. Our approach, which we call ViMoGen, tackles the generalization bottleneck by systematically unifying the strengths of high-quality motion capture (MoCap) data and the broad semantic knowledge from large-scale video generation (ViGen) models.

### 2.1 PRELIMINARIES

**Problem Definition.** Our objective is to synthesize high-quality human motion from textual descriptions, formulated as a text-to-motion translation task. Following DART (Zhao et al., 2025), we employ an overparameterized representation based on the SMPL-X (Pavlakos et al., 2019) model, where each frame is represented as a $D = 276$ dimensional vector encompassing body root translation, root rotation, joint rotations, joint locations, and joint velocities.

**Flow Matching for Motion Generation.** We employ continuous normalizing flows (Lipman et al., 2022) to model the data generation process. Given a clean motion sequence $x_0 \in \mathbb{R}^{N \times D}$, random noise $\epsilon \sim \mathcal{N}(0, I)$, and timestep $t \in [0, 1]$, we define the forward process using rectified flow interpolation: $x_t = (1 - t)\epsilon + tx_0$. The velocity field $v_t = x_0 - \epsilon$ represents the optimal transport path. Our model $f_\theta$ is trained to predict this velocity field given the noisy motion sequence $x_t$, timestep $t$, and conditioning information $c$ which represents the textual description of the desired motion. The training objective is formulated as: $\mathcal{L} = \mathbb{E}_{x_0, \epsilon, t, c}[\|f_\theta(x_t, t, c) - v_t\|_2^2]$, where $c$ encodes semantic guidance for generating motions that align with the given text prompt.

### 2.2 UNIFYING VIDEO AND MOTION GENERATION MODEL PRIOR

The fundamental challenge in motion generation lies in reconciling two competing objectives: achieving high motion quality and maintaining broad generalization capability. Traditional optical MoCap datasets provide precise motion dynamics but suffer from limited semantic diversity due

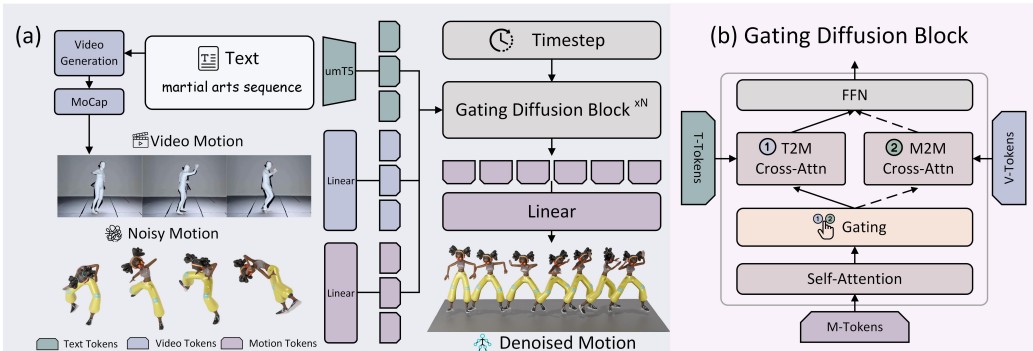

Figure 2: Overview of **ViMoGen**. (a) Our model takes a text prompt as input and leverages both a text encoder and an offline video generation model to produce textual and video motion tokens. These are fused with noisy motion inputs through a stack of gating Diffusion Blocks. (b) Each block includes self-attention, an adaptive gating module, and two cross-attention branches: Text-to-Motion (T2M) and Motion-to-Motion (M2M). Only one branch is activated at a time, enabling the model to adaptively balance robustness and generalization.

to studio constraints. Conversely, video generation models trained on web-scale data demonstrate remarkable generalization across diverse actions but produce motions with reduced fidelity and amplitude accuracy. Our approach bridges this gap through a novel cross-modal fusion architecture that adaptively leverages the complementary strengths of both modalities.

**Pipeline Overview.** As illustrated in Fig. 2(a), ViMoGen processes textual descriptions through three parallel pathways to extract complementary representations. The text encoder produces semantic embeddings $z_{text}$ that capture linguistic structure and action semantics. During inference, an offline video generation model produces videos from the text, from which we extract reference motions using visual MoCap models (Patel & Black, 2025). These are encoded into video motion tokens $z_{video}$. During training, to bridge the domain gap and improve efficiency, we simulate $z_{video}$ by perturbing ground-truth motions with controlled noise patterns that mimic visual MoCap errors. Interestingly, we find that using intermediate ViGen features as $z_{video}$ leads to slow convergence and minimal performance gains. This may be attributed to the substantial modality gap between videos and motions that is difficult to bridge with current data scales. The noisy motion tokens $z_{motion}$, text embeddings $z_{text}$, and video motion tokens $z_{video}$ are then processed through a stack of specialized cross-modal fusion blocks.

**Dual-Branch Fusion Mechanism.** Each fusion block, depicted in Fig. 2(b), employs a dual-branch architecture with mutually exclusive activation. Inspired by modern designs from WanVideo (Wang et al., 2025), our architecture begins with standard self-attention over motion tokens, followed by two specialized cross-attention branches. Notably, the Text-to-Motion (T2M) and Motion-to-Motion (M2M) branches share approximately 66% of the DiT parameters (including self-attention and FFN layers), separating only at the cross-attention mechanism. The T2M branch enables motion tokens to attend directly to text embeddings, leveraging the precise motion dynamics learned from high-quality optical MoCap data. This branch performs well in generating physically plausible motions for actions well-represented in traditional mocap datasets. The M2M branch allows motion tokens to attend to video motion tokens, transferring semantic knowledge from the video generation domain. This branch captures the rich contextual understanding and generalization capabilities derived from web-scale video training data, enabling the model to handle novel or rare actions absent from MoCap datasets. The key insight is that these branches operate in complementary regimes: the T2M branch provides robustness and quality assurance, while the M2M branch extends semantic coverage and generalization.

**Adaptive Branch Selection.** The selection between branches is governed by an adaptive mechanism that assesses the semantic alignment between the text prompt and the generated video motion. During inference, we first generate video motion offline using the text prompt, then employ a Vision-Language Model to perform a binary alignment check between the generated video and the text prompt. If aligned, the M2M branch is activated to refine the video motion; otherwise, the model falls back to the T2M branch to generate motion from text. This instance-level adaptive gating strategy enables ViMoGen to dynamically balance between leveraging video model generalization for novel actions and maintaining motion quality through established mocap priors. During training,

we employ a curriculum approach based on dataset characteristics to regulate branch activation. For manually annotated datasets, we prioritize the T2M branch to ensure precise semantic alignment; for large-scale auto-labeled datasets, we encourage broader generalization by increasing M2M branch usage. When the M2M branch is activated, we generate $z_{video}$ by applying a compound noise strategy to ground-truth motions—including random corruption, jitter simulation, and temporal dropout—to mimic the artifacts of visual MoCap. Furthermore, since visual MoCap often yields inaccurate global trajectories, we mask the global translation in $z_{video}$, forcing the M2M branch to rely on local pose dynamics. This simulation avoids the need for online video generation during training.

## 2.3 DISTILLING MOTION PRIOR FROM VIDEO GENERATION MODEL

While ViMoGen unifies video generation generalization with motion generation quality, the requirement for video generation model inference introduces significant computational overhead. To address this limitation, we propose an efficient variant, ViMoGen-light, which exclusively uses the T2M branch and eliminates video generation dependencies.

**Semantically-Diverse Data Synthesis.** Our preliminary experiments reveal that model generalization is closely correlated with the diversity of action verbs and phrases encountered during training (detailed in Section 5). Based on this key insight, we systematically construct a comprehensive set of action descriptions to capture the full spectrum of human motion semantics. We begin by extracting high-frequency verbs and action phrases from linguistic resources, followed by semantic clustering and expansion using large language models (LLMs). This process yields a comprehensive synthetic dataset of 14,000 novel prompts covering diverse action categories, such as *a person is breakdancing* and *a knight is jousting*, which are crucial for enhancing generalization beyond standard benchmarks.

**Knowledge Distillation.** The ViMoGen-light model is created through knowledge distillation, where the generalization capabilities of the T2V prior are *distilled* into a lightweight architecture. The complete ViMoGen pipeline serves as a powerful teacher model. For each of the synthesized prompts, the teacher generates a corresponding high-quality motion. The student model, ViMoGen-light, is then trained on this synthetic dataset using standard flow matching objectives. Through this process, the student model inherits broad semantic knowledge and generalization priors from the teacher's dual-branch architecture, thereby enabling inference without a ViGen model. Ablation studies confirm that this distilled data is highly effective for improving motion generation model generalization while maintaining computational efficiency.

## 3 VIMOGEN-228K DATASET

In this section, we introduce **ViMoGen-228K**, a dataset carefully designed from a broad spectrum of sources, featuring a competitive scale, high motion quality and broad semantic diversity. Table 1 presents a comparison between ViMoGen-228K and existing datasets. Optical MoCap datasets (*e.g.*, HumanML3D (Guo et al., 2022a)) are indispensable for establishing strong motion priors but they are limited in scale. In contrast, web video–based datasets scale far more easily but typically compromise motion quality and often exhibit semantic biases (e.g., Motion-X (Lin et al., 2023)

| Dataset | #Clip | #Hour | Scene | Motion | Video |
|---|---|---|---|---|---|
| KIT-ML | 3911 | 11.2 | Indoor | GT | - |
| AMASS | 11,265 | 40 | Indoor | GT | - |
| BABEL | 13,220 | 43.5 | Indoor | GT | - |
| HumanML3D | 14,616 | 28.6 | Indoor | GT | - |
| Motion-X | 81,084 | 144.2 | In-the-Wild | Pseudo GT | ✓ |
| Motion-X++ | 120,500 | 350 | In-the-Wild | Pseudo GT | ✓ |
| MotionMillion | 2M | >2000 | In-the-Wild | Pseudo GT | ✓ |
| ViMoGen-228K | 228,236 | 369.4 | | | |
| • Optical MoCap[†] | 171,542 | 292.7 | Indoor | GT | - |
| • In-the-Wild Video[‡] | 41,971 | 61.4 | In-the-Wild | Pseudo GT | ✓ |
| • Synthetic Video[#] | 14,723 | 16.6 | In-the-Wild | Pseudo GT | ✓ |

Table 1: Comparison of **ViMoGen-228K** with existing human motion datasets. [†]Unified 29 datasets. [‡]Aggressively filtered from 10M clips. [#]Strategically generated for semantic coverage.

focuses predominantly on sports and gaming scenarios). ViMoGen-228K is designed to strike a deliberate balance between motion quality (172K high-fidelity text-motion pairs) and semantic diversity (56K diverse text-video-motion triplets) through three complementary strategies: (1) aggregating a large collection of 30 optical MoCap datasets, surpassing the scale of any previous optical MoCap resource; (2) filtering a massive pool of web videos through a rigorous selection pipeline to maximize motion fidelity while enriching in-the-wild representations; and (3) strategically generating synthetic videos to further expand semantic coverage, particularly in domains underrepresented or difficult to capture in real-world datasets.

We explain the data collection, filtering, and annotation below. Due to space limitations, more details are available in Appendix D.2.

**Optical MoCap Dataset.** We unify 30 publicly available datasets (for the complete list of datasets, please refer to Appendix D.2.1), standardizing to SMPL-X (Pavlakos et al., 2019) format and 20 frames-per-second (fps). Long sequences are segmented into 5-second clips (100 frames) without overlap. Multi-stage filtering includes T-pose removal, minimum 3-second length requirements, and quality assessment via MBench to eliminate jitter and low-dynamics motions. Empirical analysis led us to exclude several datasets due to poor text-motion consistency and challenging high-speed motion characteristics in actions like dancing. Eventually we select 17 of the 30 candidate datasets (172k of 267k clips) for our final training corpus. For existing MoCap datasets like HumanML3D, we utilize their original text annotations when available. For optical MoCap datasets that lack text annotations or only have class labels, we design a pipeline to generate structured textual descriptions. Specifically, we render depth videos of the human mesh for each motion sequence, then provid frames from these videos at 5 fps to Gemini 2.0 for motion description. We engineer a system prompt to elicit structured annotations, including a brief summary, key actions, frame-level details, and motion style. We apply heuristic filtering to multimodal LLM-generated annotations, removing invalid responses.

**In-the-Wild Video Data.** From an internal mega-scale video database (consisting of large public datasets (Wang et al., 2024b; Chen et al., 2024; Ju et al., 2024; Grauman et al., 2024)), and videos obtained via keyword searches across sports, daily activities, cultural performances, and occupational tasks), we obtain 10M human-centric video clips after quality assessment (removing clips with low visual quality, shaky camera, excessive motion, poor lighting, or heavy occlusion), making the source competitive in scale against existing works (Lu et al., 2025; Fan et al., 2025). We annotate the clips with CameraHMR (Patel & Black, 2025) and SMPLest-X (Yin et al., 2025) to extract 3D motion in the SMPL-X representation. We then canonicalize the global orientation of each motion so that the initial frame faces the y+ axis. Visual MoCap artifacts, such as jitter and inaccurate global translation, are mitigated by applying post-processing smoothness algorithms. Furthermore, only local body poses are supervised during training. For video clips without text annotation, we provide the original RGB videos as input to Gemini 2.0 Flash for motion captioning.

**Synthetic Video Data.** With the rapid development in ViGen, leading models (Wan et al., 2025) trained on billions of images and videos demonstrate unprecedented controllability and generalization capability. Leveraging these characteristics, we explicitly instruct the ViGen model to generate videos that are easy to perform vision-based MoCap (*e.g.*, stable camera movement, full body, single person) that are difficult to find the equivalent in real-world videos. To systematically expand semantic coverage into underrepresented domains, we construct a long-tail vocabulary derived from large-scale video captions in ViGen datasets such as OpenHumanVid (Li et al., 2024), LLaVA-Video-178K (Zhang et al., 2024c), and an internal collection. The compiled action verbs and descriptive nouns undergo semantic clustering and deduplication, yielding over 20,000 diverse prompts, w. These prompts are then processed with the Wan2.1 (Wan et al., 2025) text-to-video model to produce 81-frame clips at 16 frames per second (fps). The results further undergo standard filtering and annotation pipeline with additional refinement via our pre-trained ViMoGen M2M branch to obtain 14,000 high-quality samples.

## 4 MBench

In this section, we introduce the main components of MBench, a novel hierarchical benchmark designed to provide a granular and multifaceted assessment of generated human motions. Section 4.1 outlines the motivation behind the design of the 9 evaluation dimensions, along with their definitions and evaluation methods. To validate MBench 's alignment with human perception, we conduct human preference annotations for each evaluation dimension, as discussed in Appendix C.1.

### 4.1 EVALUATION DIMENSIONS

As shown in Fig. 3(b), MBench offers a structured approach by decomposing the evaluation into nine dimensions in three primary pillars: Motion Generalization, Motion-Condition Consistency, and Motion Quality. Finally, we construct and release two curated prompt lists designed to go beyond

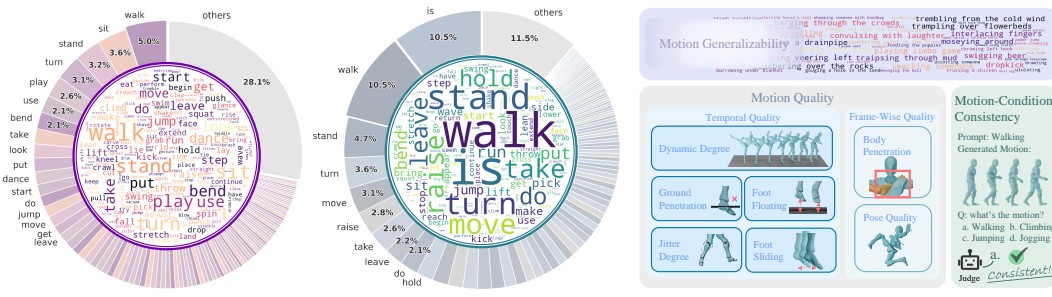

(a) Top 100 Verbs (Left: MBench. Right: HumanML3D)    (b) Evaluation Dimensions of MBench

Figure 3: Overview of **MBench**. (a) MBench features more balanced distribution and vastly different prompt designs compared to HumanML3D. (b) MBench designed is to systematically evaluate motion generation algorithms across nine dimensions, focusing on motion quality, prompt-following, and generalization capability.

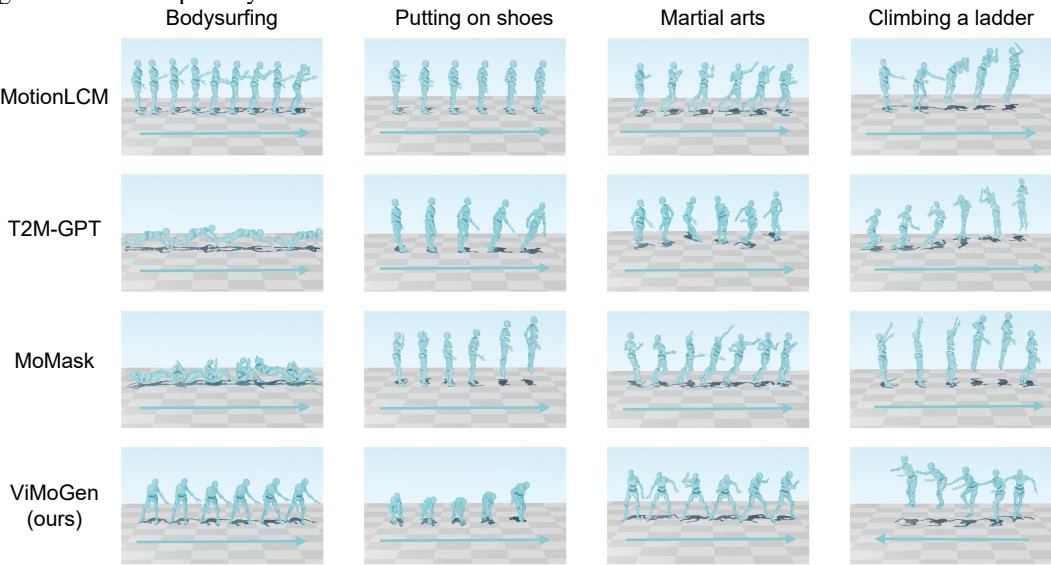

Figure 4: Qualitative comparison on MBench prompts. We show keywords in prompts for simplicity.

simple memorization and dataset overlap, serving as challenging cases for model evaluation. Details of the prompt suits are included in Appendix C.2.

**Motion Generalizability.** This dimension specifically targets motions that are rare or entirely absent from commonly used MoGen and ViGen datasets. By evaluating such out-of-distribution actions, it directly measures a model's ability to transcend memorization of seen patterns and instead generalize to novel semantic inputs. In doing so, it highlights the capacity of the model to produce diverse, contextually coherent movements. A single action (*e.g.*, *walking*) can be expressed through numerous linguistic variations that convey subtleties of emotion, speed, and intent. To construct an open-world vocabulary for this dimension, we systematically extract near-synonyms and related expressions from the Oxford English Dictionary. These are then composed into descriptive sentences that encode contextual variations in physical style and temporal dynamics. For example, the simple action *trample* is expanded into a semantically richer phrase such as *trample over flowerbeds with heavy, lumbering steps*. This methodology ensures that the benchmark probes deeper language–motion associations rather than merely testing surface-level mappings. For evaluation, generated motions are rendered as videos with realistic ground planes and natural camera movements, thereby avoiding oversimplified stick-figure visualizations. A vision–language model (VLM) is then tasked with assessing whether the generated motions faithfully depict the specified rare or unseen actions.

**Motion-Condition Consistency.** This metric quantifies semantic fidelity between generated motion and text prompts. While pre-trained action recognition models like TMR (Petrovich et al., 2023) could assess action accuracy, they suffer from training data biases and limited generalization. UMT (Li

| Models | Motion Condition Consistency ↑ | Motion Generalizability ↑ | Jitter Degree ↓ | Dynamic Degree ↑ | Foot Floating ↓ | Foot Sliding ↓ | Body Penetration ↓ | Pose Quality ↓ |
|---|---|---|---|---|---|---|---|---|
| MDM (Tevet et al., 2023) | 0.42 | 0.51 | 0.0136 | 0.0376 | 0.156 | 0.0136 | 1.68 | 2.67 |
| T2M-GPT (Zhang et al., 2023a) | 0.39 | 0.38 | 0.0156 | 0.0349 | 0.209 | 0.0156 | 1.33 | 2.43 |
| FineMoGen (Zhang et al., 2023d) | 0.37 | 0.42 | 0.0118 | 0.0386 | 0.281 | 0.0091 | 1.18 | 2.28 |
| MotionLCM (Dai et al., 2024) | 0.48 | 0.55 | 0.0218 | **0.0439** | 0.193 | 0.0202 | 1.73 | 2.40 |
| MoMask (Guo et al., 2024) | 0.38 | 0.44 | 0.0147 | 0.0396 | 0.178 | 0.0147 | 1.48 | 2.67 |
| MotionDiffuse (Zhang et al., 2024b) | 0.44 | 0.42 | 0.0111 | 0.0289 | **0.126** | 0.0063 | 1.35 | 2.21 |
| MotionCraft (Bian et al., 2025) | 0.42 | 0.45 | 0.0132 | 0.0420 | 0.402 | 0.0090 | **1.15** | 2.12 |
| ViMoGen (Ours) | **0.53** | **0.68** | **0.0108** | 0.0251 | 0.204 | 0.0064 | 1.78 | 2.38 |
| ViMoGen-light (Ours) | 0.47 | 0.55 | 0.0129 | 0.0294 | 0.155 | **0.0051** | 1.43 | **2.10** |

Table 2: Quantitative comparison on MBench. The best performance is bolded.

et al., 2023b), used in VBench (Huang et al., 2024a;b), evaluates action execution via video-text alignment but underperforms in motion-only scenarios lacking rich visual context. For this assessment, we adopted an evaluation pipeline similar to that used for Motion Generalizability. Using chain-of-thought prompting, the VLM describes observed motion, then selects the most suitable label from ten candidates: one ground-truth and nine distractors. Distractors are formed using cosine similarity thresholds from three percentile ranges (below 5th, 47-52nd, above 95th), ensuring semantic variability and consistent evaluation.

**Motion Quality.** Motion Quality evaluation is decomposed into temporal and frame-wise aspects. *Temporal Quality* assesses cross-frame consistency: jitter degree (quantifying articulatory instability via joint accelerations and orientation changes), foot contact metrics including ground penetration, foot floating, and foot sliding (evaluating physical plausibility of foot-ground interactions), and dynamic degree (measuring overall motion intensity through joint velocities). *Frame-Wise Quality* evaluates individual pose characteristics, including body penetration rates computed via BVH-based collision detection and pose naturalness assessed using a Neural Riemannian Distance Field (NRDF) model (He et al., 2024) that captures anthropomorphic plausibility priors.

# 5 EXPERIMENTS

In this section, we compare ViMoGen with state-of-the-art (SOTA) methods on MBench (Section 5.1), and conduct ablation studies (Section 5.2). We include implementation details (Section E) and results (Section F) on the standard HumanML3D benchmark (Guo et al., 2022a) in the Appendix.

## 5.1 COMPARISON WITH SOTA METHODS

We benchmark ViMoGen and its distilled variant, ViMoGen-light, against leading motion generation models: MDM (Tevet et al., 2023), MotionLCM (Dai et al., 2024), T2M-GPT (Zhang et al., 2023a), MotionDiffuse (Zhang et al., 2024b), FineMoGen (Zhang et al., 2023d), MotionCraft (Bian et al., 2025), and MoMask (Guo et al., 2024) using our comprehensive MBench evaluation framework.

**Quantitative Results.** Table 2 presents the quantitative comparison on MBench. Our full model, ViMoGen, significantly outperforms all baselines on the key semantic metrics of Motion Condition Consistency and Generalizability, showcasing the powerful advantage of leveraging a T2V model's rich semantic prior. The distilled ViMoGen-light variant achieves a Generalization Score on par with the strongest baseline, demonstrating that this knowledge can be effectively transferred to an efficient model that does not require video generation at inference. Our improved semantic performance stems from using a more powerful T5 text encoder and a multi-token cross-attention mechanism, which provides richer textual guidance than the single-token CLIP features used in prior works. This pursuit of generalization introduces a trade-off in motion quality. By incorporating diverse, video-sourced motions (e.g., "tying shoelaces"), our training data features more complex actions with less global movement compared to locomotion-heavy datasets. This data characteristic leads our models to produce more stable patterns, explaining both their superior Jitter Degree and lower Dynamic Degree.

**Qualitative Analysis.** Fig. 4 presents qualitative results using representative prompts of each MBench dimension. ViMoGen demonstrates superior text-to-motion alignment and physical plausibility across diverse scenarios. For out-of-domain prompts like 'body surfing', ViMoGen leverages semantic knowledge from video generation priors to produce plausible motions despite this action being absent

| Branch Selection | Motion Condition Consistency | Motion Generalizability | Jitter Degree | Foot Sliding |
|---|---|---|---|---|
| Video Generation Baseline | 0.51 | 0.58 | 0.0193 | 0.0161 |
| T2M Only | 0.46 | 0.54 | 0.0111 | **0.0039** |
| M2M Only | 0.51 | 0.59 | 0.0145 | 0.0113 |
| Adaptive Gating | **0.53** | **0.68** | **0.0108** | 0.0064 |

| Training Text Style | Testing Text Style | Motion Condition Consistency | Motion Generalizability | Foot Sliding |
|---|---|---|---|---|
| Motion | Motion | 0.36 | 0.40 | 0.0032 |
| Motion | Video | 0.32 | 0.39 | **0.0031** |
| Video | Motion | **0.43** | **0.48** | 0.0033 |
| Video | Video | 0.41 | 0.44 | 0.0032 |

Table 3: Ablation study on different branch selection strategies for ViMoGen. Our adaptive method significantly outperforms single-branch baselines in generalization and accuracy.

Table 4: Ablation on text prompt style. Training with descriptive *video-style* text and testing on concise *motion-style* text yields the best overall performance.

| Training Datasets | Motion Clip Number | Motion Condition Consistency | Motion Generalizability | Foot Sliding |
|---|---|---|---|---|
| HumanML3D | 89k | 0.41 | 0.44 | **0.0032** |
| + Other Optical Mocap Data | 83k | 0.44 | 0.48 | 0.0033 |
| + Visual Mocap Data | 42k | 0.43 | 0.50 | 0.0042 |
| + Synthetic Video Data | 14k | **0.47** | **0.55** | 0.0051 |

| Text Encoder | Motion Condition Consistency | Motion Generalizability | Foot Sliding | Body Penetration |
|---|---|---|---|---|
| CLIP | 0.32 | 0.35 | **0.0023** | 1.39 |
| T5-XXL | **0.41** | 0.44 | 0.0032 | **1.05** |
| MLLM | 0.38 | **0.46** | 0.0032 | 1.51 |

Table 5: Ablation on the composition of the training data. Adding diverse data sources progressively improves generalization, with synthetic data providing the largest gains.

Table 6: Ablation on the choice of text encoder. Compared to CLIP (Radford et al., 2021) and MLLM (Kong et al., 2024), T5-XXL (Raffel et al., 2020) provides the best balance of generalization and motion quality.

from standard training datasets. Competing methods such as T2M-GPT (Zhang et al., 2023a) generate implausible or generic motions for such novel prompts. ViMoGen also excels on common daily actions, confirming robust performance across the full semantic spectrum. These results validate our core contribution of bridging the semantic gap between limited mocap data and diverse real-world motion requirements.

## 5.2 ABLATION STUDY

**Effectiveness of Adaptive Selection.** We validate our adaptive branch selection architecture by comparing against several baselines in Table 3. The video generation baseline, using motion estimated directly from text-to-video models, produces semantically relevant but low-quality motions with significant jitter and foot sliding. The M2M model uses this noisy video-based motion as a reference, improving quality metrics but without significant gains in accuracy or generalization. Conversely, the T2M model, trained solely on high-quality MoCap data, achieves optimal motion quality but exhibits limited generalization capabilities. Our adaptive gated model successfully integrates the strengths of both approaches. The source of this improved generalization lies in its intelligent fallback mechanism. For novel prompts where the ViGen prior is semantically accurate, our model leverages its rich guidance. However, for highly dynamic actions like turning around or falling, where ViGen models can be unstable and produce distorted motions (e.g., "body twisting"), our adaptive mechanism switches to the more robust T2M branch. This branch, guided by the powerful semantic priors of the text encoder, ensures plausible motion generation when the video-based guidance is unreliable. We further provide qualitative examples illustrating this adaptive process in Appendix G.

**Impact of Multi-Source Training Data.** We analyze the contribution of each data component within our ViMoGen-228K dataset by incrementally adding data sources to a baseline ViMoGen-light model trained solely on HumanML3D (Guo et al., 2022a). The results in Table 5 show that progressively adding data from diverse sources steadily improves the model's action accuracy and generalization. Notably, the inclusion of synthetic video data, despite its small size (14k clips), provides a substantial boost to the generalization score (from 0.50 to 0.55). This progression confirms that semantic diversity, even from smaller datasets, significantly impacts generalization.

**Impact of Text Encoder and Representation.** We investigate how different text encoders and representations affect model performance. First, we compare three pre-trained text encoders: CLIP (Radford et al., 2021), T5-XXL (Raffel et al., 2020), and a Multimodal Large Language Model (MLLM) (Kong et al., 2024). As shown in Table 6, both T5-XXL and the MLLM significantly outperform CLIP in generalization capacity. While prior works like MoMask (Guo et al., 2024) successfully employed CLIP on the HumanML3D (Guo et al., 2022a) benchmark, our findings indicate that more powerful text encoders are essential for handling the greater linguistic complexity

| T2M Branch | M2M Branch | Motion Condition Consistency ↑ | Motion Generalizability ↑ | Jitter Degree ↓ | Dynamic Degree ↑ | Foot Floating ↓ | Foot Sliding ↓ | Body Penetration ↓ | Pose Quality ↓ |
|---|---|---|---|---|---|---|---|---|---|
| **ViMoGen (Ours)** | **ViMoGen (Ours)** | **0.53** | **0.68** | **0.0108** | 0.0251 | **0.204** | **0.0064** | 1.775 | **2.382** |
| ViMoGen (Ours) | DNO | 0.50 | 0.65 | 0.0138 | 0.0320 | 0.255 | 0.0075 | **1.725** | 2.453 |
| MotionLCM | ViMoGen (Ours) | 0.49 | 0.66 | 0.0179 | 0.0374 | 0.217 | 0.0136 | 1.870 | 2.483 |
| MotionLCM | DNO | 0.49 | 0.63 | 0.0205 | **0.0432** | 0.297 | 0.0217 | 1.984 | 2.567 |

Table 7: Analysis of mutual benefits between T2M and M2M branches. We compare our joint training approach against baselines using MotionLCM (Dai et al., 2024) and DNO (Karunratanakul et al., 2024).

required for true generalization. Next, we analyze the impact of prompt style, comparing concise *motion-style* text with rich, descriptive *video-style* text. The detailed prompts and visualization examples are available in Appendix G. Table 4 reveals that training with descriptive *video-style* text while testing on concise *motion-style* descriptions yields the best performance. This suggests rich descriptions function as effective data augmentation, improving model robustness and alignment with pretrained text encoder expectations.

**Mutual Benefits of Dual-Branch Training.** To clarify whether the T2M and M2M branches benefit each other, we analyze our joint training strategy where branches share approximately 66% of DiT parameters (self-attention and FFN layers). In Table 7, we compare our unified approach against a selection-based baseline that uses external VLM scores to choose between a SOTA T2M model (MotionLCM (Dai et al., 2024)) and a motion refinement pipeline (DNO (Karunratanakul et al., 2024)). We also perform ablations replacing one of our branches with these baselines. The results demonstrate that our joint training significantly outperforms simple model selection and single-branch hybrids. The M2M branch introduces diverse motion priors from optical MoCap data (often lacking text labels) into the shared parameters, enhancing the plausibility of T2M generation. Conversely, the T2M branch enforces semantic alignment, benefiting the overall representation. This synergy results in superior condition consistency and generalizability compared to using separate SOTA models.

## 6 CONCLUSION AND DISCUSSION

In this work, we tackle the fundamental challenge of generalizable 3D human motion generation through coordinated innovations in data, modeling, and evaluation. (1) **ViMoGen-228K** dataset is curated with more than 228,000 motion clips featuring both high-quality and broad semantic coverage. (2) **ViMoGen** model unifies priors from video generation with motion-specific knowledge through innovative gated diffusion blocks, achieving state-of-the-art performance in both action accuracy and generalization. Additionally, the lightweight variant, **ViMoGen-light**, effectively distills generalization capabilities while significantly reducing computational overhead. (3) **MBench** provides the first comprehensive benchmark for fine-grained evaluation across generalization capabilities, motion-condition consistency motion quality. Moreover, a detailed discussion on related work is included in Appendix B.

**Limitations and Future Work.** Despite these advancements, our method currently faces two primary limitations. First, the architecture is designed for single-person motion generation and does not yet support multi-person interactions. Second, for complex, high-dynamic motions (e.g., gymnastic twists), the model relies heavily on the video generation model for initialization; if the video prior is distorted, the M2M branch may struggle to fully correct the dynamics. We also observe a trade-off where superior generalization does not always yield the highest scores on specific quality metrics (e.g., Dynamic Degree). This is primarily attributed to artifacts in current visual MoCap data (such as foot sliding) and the limited dynamic range inherent in generated videos. Future work will address these challenges by incorporating advanced contact-aware MoCap algorithms to enhance data precision and exploring training strategies that force the model to extrapolate high-quality dynamics from video initializations.

## ETHICS STATEMENT

This work adheres to the ICLR Code of Ethics. Our study relies on motion datasets that may contain data derived from human subjects. This may raise potential concerns regarding privacy and consent. To mitigate these issues, we only use publicly available optical motion capture datasets released under research-friendly licenses. For video data collected from the internet, we focus solely on motion information, and no personally identifiable information was used in our study. Regarding the motion generation model, while it has the potential for positive applications in animation, robotics, and rehabilitation, it could also be misused for deceptive content creation or surveillance. We acknowledge this risk and emphasize that our model is intended solely for scientific research and beneficial applications.

## REPRODUCIBILITY STATEMENT

We have taken extensive steps to ensure the reproducibility of our work. All code, datasets, and benchmark configurations will be made publicly available in an anonymous repository. Details of data processing, experimental setup, and training protocols are thoroughly described in the main paper and supplementary materials. Hyperparameters, implementation choices, and evaluation metrics are also documented to enable independent verification and fair comparison.

## ACKNOWLEDGEMENTS

This research is supported by cash and in-kind funding from NTU S-Lab and industry partner(s). This study is also supported by the Ministry of Education, Singapore, under its MOE AcRF Tier 2 (MOE-T2EP20221-0012, MOE-T2EP20223-0002).

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

## A  AI USAGE DECLARATION

Artificial intelligence tools (e.g., ChatGPT) were utilized solely for the purposes of grammar checking and language refinement. No content was generated that contributed to the intellectual or analytical aspects of this work.

## B  RELATED WORK

### B.1  MOTION DATASET

**Optical MoCap Dataset.** Pioneering works in text-to-motion generation relied almost exclusively on optical MoCap datasets. These resources, exemplified by KIT-ML (Plappert et al., 2016), AMASS (Mahmood et al., 2019), BABEL (Punnakkal et al., 2021),HumanML3D (Guo et al., 2022a), aioz (Le et al., 2023), etc, were instrumental in advancing the field. Optical motion capture provides precise, low-noise joint position and rotation data, resulting in motions that are physically plausible and temporally coherent. This precision is critical for modeling the complex dynamics of the human body, such as maintaining balance and ground contact. The controlled, studio-based collection environment also ensures that the data is meticulously clean and free from common artifacts like jitter or body-part penetration. Despite these strengths, these datasets are orders of magnitude smaller than those in other modalities. HumanML3D, for example, contains approximately 14,000 clips. This scarcity is a direct consequence of the labor-intensive and expensive collection process. The controlled environment and small scale also lead to a severely limited semantic coverage, failing to capture the broad distribution of human movements. This limitation means that models trained solely on these datasets tend to overfit to a small set of indoor, locomotion-heavy actions, thereby losing the ability to generalize to novel instructions. This bottleneck has been a primary driver for the field's search for alternative data sources.

**Visual MoCap Datasets.** To break free from the MoCap bottleneck, the community has turned to large-scale, video-based data. These "in-the-wild" datasets leverage advancements in visual motion capture (visual MoCap) to extract motion from public web videos. Motion-X (Lin et al., 2023) was an early step in this direction, compiling approximately 100,000 sequences, but it were still considered limited in scale and primarily focused on specific scenarios such as sports and gaming. The MotionMillion (Fan et al., 2025) dataset represents the pinnacle of this approach, with over 2 million high-quality motion sequences and 2,000 hours of content, making it 20 times larger than existing resources. While immensely valuable, this approach introduces a new set of challenges. The motion data is fundamentally derived from the visual mocap method, which has lower fidelity manifests as kinematic artifacts and jitter. This transition reveals a critical shift: the problem of data scarcity has been replaced by the problem of data quality heterogeneity. New methods must be developed not just to collect data, but to effectively handle and learn from data of varying fidelity.

### B.2  TEXT-TO-MOTION GENERATION MODEL

**Conventional Motion Generation Models.** The current state of the art has been overwhelmingly driven by two paradigms: diffusion and autoregressive models. Diffusion models like MDM (Tevet et al., 2023), MotionDiffuse (Zhang et al., 2024b), and MotionLCM (Dai et al., 2024) have become the dominant paradigm for generating high-fidelity motion. They are praised for their ability to synthesize natural motions by denoising a random signal over time. More recently, ReMoDiffuse (Zhang et al., 2023c) proposed a retrieval-augmented diffusion model to address the unsatisfactory performance of models on diverse motions. However, unlike models that leverage knowledge from other modalities like video generation, ReMoDiffuse's approach is based on retrieving and refining from a database of existing motion samples. Autoregressive approaches like T2M-GPT (Zhang et al., 2023a) and ScaMo (Lu et al., 2025) model motion as a sequence of discrete tokens. This approach, akin to Large Language Models (LLMs), is inherently scalable and well-suited for modeling long, compositional sequences. However, a critical limitation shared by these models is their reliance on constrained datasets. While they achieve impressive results on benchmarks like HumanML3D, their generalization to out-of-domain instructions remains a challenge.

**Bridging Motion and Video Generation Priors.** A new class of models is emerging to address this limitation by leveraging the vast semantic knowledge encoded in large-scale video generation

models. Early works (Millán et al., 2025; Albaba et al., 2025) attempted to leverage knowledge from large-scale video generation models by animating images and then extracting motion through optimization-based methods. The key critique of these methods is their heavy dependence on video generation model performance, which often lacks robust motion quality; their neglect of existing motion datasets that provide valuable human motion priors and patterns; and high computational costs and slow inference speeds.

## C  MBench details

### C.1  Human Preference Analysis

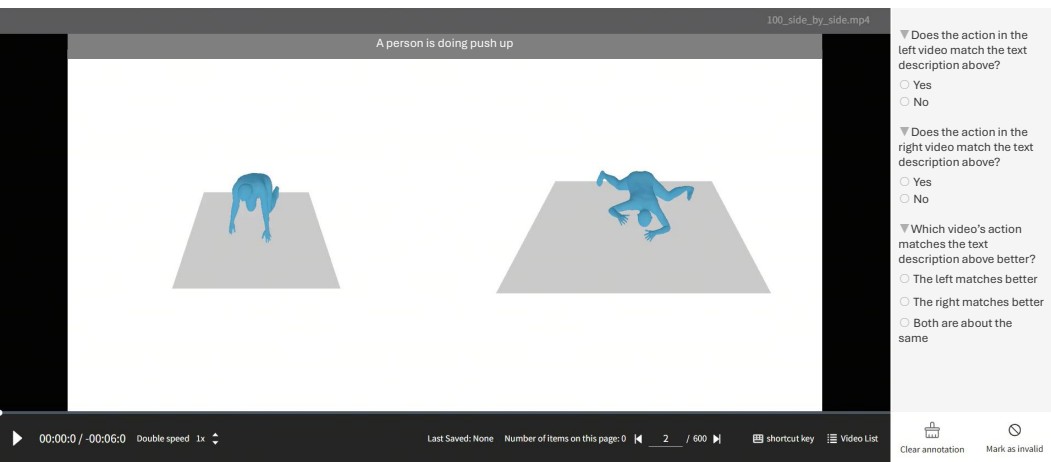

Figure 5: **Human Preference Annotation Interface.** Two rendered videos side-by-side with annotator choices.

To validate MBench 's alignment with human perception across the nine evaluation dimensions, we conducted large-scale human preference labeling on rendered motion videos. The annotators range in age from 20 to 35 and are equipped with fundamental domain knowledge relevant to the task.

#### C.1.1  Data Preparation

For each text prompt $p_i$, we generate videos using five text-to-motion models $\mathcal{M} = \{M_1, M_2, M_3, M_4, M_5\}$, producing outputs $G_i = \{V_{i,1}, V_{i,2}, V_{i,3}, V_{i,4}, V_{i,5}\}$. We construct all pairwise combinations within each group, yielding $\binom{5}{2} = 10$ unique pairs per prompt.

Different evaluation protocols target specific metrics. For Motion-Condition Consistency and Motion Generalizability, annotators perform pairwise comparisons to determine which video better satisfies the text prompt. For physical and perceptual quality metrics (Jitter Degree, Ground Penetration, etc.), annotators rate each video individually using a 3-point Likert scale (0-2) with natural language descriptions. Each video appears 5 times during evaluation with randomized presentation order to minimize bias.

Across $N$ prompts, this yields $N \times 10$ pairwise comparisons. Aggregated results quantify the correlation between MBench automatic metrics and human judgments, enabling rigorous assessment of metric reliability and model performance.

#### C.1.2  Validating Human Alignment of MBench

Given the human annotation results, we compute the win ratio for each model. In each pairwise comparison, if a model's video is judged as better, the model receives a score of 1 while the other receives 0. In the case of a tie, both models are assigned a score of 0.5.

**Per-Dimension Evaluation.**For each evaluation dimension, we compute the win ratio using two sources: (1) scores derived from MBench 's automatic evaluation metrics, and (2) human annotation

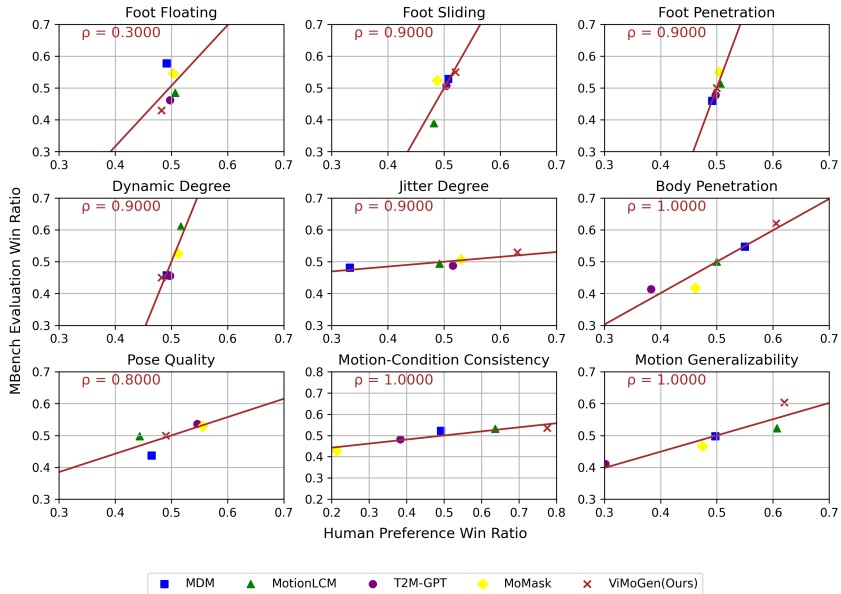

Figure 6: **MBench's Human Alignment.** In each plot, a dot represents the human preference win ratio (horizontal axis) and MBench automatic evaluation win ratio (vertical axis) for a motion generation model. We linearly fit a straight line to visualize the correlation and calculate the Spearman's correlation coefficient ($\rho$) for each dimension.

results. We then calculate the correlation between these two sets of win ratios. As illustrated in Fig. 6, the scatter plots show strong positive correlations between human-preference win ratios and MBench per-dimension evaluation win ratios, indicating the reliability of the proposed automatic metrics. Notably, the Foot Floating dimension has a lower correlation. This is because human generally have limited sensitivity to subtle foot-floating artifacts, causing their preference scores to cluster around 0.5. In contrast, the physics-based metric in our benchmark can precisely distinguish these differences. We acknowledge the slight difference in calculating the MBench evaluation win ratio for Motion Generalizability and Motion–Condition Consistency. As mentioned in Sec. 4.1, these two dimensions are assessed through multiple-choice tasks, where the VLM first describes the motion and then selects the most matching text description from 10 carefully selected text options. The selection is correct, the sample scores; otherwise it doesn't. This results in binary correctness rather than continuous [0,1] scores. Consequently, when computing MBench evaluation win ratios, two motion generation methods that both receive either 0 or 1 for a sample result in a tie, which reduces the variation in their MBench win ratio distribution.

**Usage of Single-Video Evaluations for Disambiguation**. In addition to pairwise comparisons, annotators also provide single-video ratings for each video from each method. These individual scores are used to mitigate potential biases or unfairness in pairwise annotations. Specifically, in cases where the single-video scores clearly indicate that one video is superior but the corresponding pairwise annotation contradicts this assessment, the decision is revised in favour of the single-video judgment.

## C.2    PROMPT SUITE PER MBENCH EVALUATION DIMENSION

The prompts used in our benchmark are carefully curated to balance evaluation efficiency with representativeness. We aim to avoid excessive number of prompts that could lead to a long evaluation time. While we still need to ensure the suite is comprehensive enough to cover diverse motion types and possesses the discriminative power to reveal differences among evaluation dimensions. In total, we designed a suite of 450 distinct prompts, each tailored to a specific MBench dimension:

- 150 prompts for Temporal Quality(**Motion Quality**)
- 100 prompts for Frame-Wise Quality(**Motion Quality**)

- 100 prompts for **Motion-Condition Consistency**

- 100 prompts for **Motion Generalizability**

### C.2.1 MOTION QUALITY PROMPTS

For the Temporal and Frame-Wise Quality dimensions, the primary goal is to assess the intrinsic quality of the generated motion, rather than the model's ability to simply produce an output. Thus, these 250 prompts were sourced from established benchmarks, such as HumanML3D (Guo et al., 2022a) and AMASS (Mahmood et al., 2019), as well as various open-source motion libraries. The final test set was manually curated to specifically include challenging scenarios where current models are known to fail, such as those prone to artifacts like foot-skating, ensuring a rigorous evaluation of motion fidelity.

### C.2.2 MOTION-CONDITION CONSISTENCY PROMPTS

The prompts for Motion-Condition Consistency is designed to probe robustness of motion generation models. The prompts intentionally also include common, everyday actions that are typically under-represented in traditional motion generation datasets, testing the model's ability to generalize beyond its training data.

The creation process for this prompt suite was as follows:

Data Sourcing: We drew from a diverse range of actions found in large-scale video generation datasets, including LLaVA-178K (Zhang et al., 2024c) and an extensive internal dataset containing millions of video-text pairs.

Automated Extraction: The DeepSeek-R1 (Guo et al., 2025) model was utilized to parse and extract motion-specific descriptions from the video annotations and text prompts.

Semantic Embedding and Clustering: After removing duplicates, we mapped the textual descriptions into a shared embedding space using several encoders, including T5 (Raffel et al., 2020), CLIP (Radford et al., 2021), and BGE-M3 (Multi-Granularity, 2024). BGE-M3 was ultimately selected for its superior semantic clustering performance. We then applied K-means clustering to the resulting embeddings, generating 1000 distinct prompt clusters.

Final Curation: The final set of prompts was selected through a hybrid approach, combining automated filtering with DeepSeek-R1 and expert curation to ensure relevance, diversity, and difficulty.

### C.2.3 MOTION GENERALIZABILITY

The prompts are designed to evaluate a model's generalization capacity by requiring it to handle motions described by an open-world vocabulary, rather than just the limited terms found in standard datasets. A single action, such as walking, can be expressed with numerous linguistic nuances that convey subtleties in emotion, speed, and intent. A good generalized model can translate subtle semantic differences into distinct kinematic profiles, proving it possesses a genuine understanding of the relationship between language and motion.

For prompt creation, our methodology involved the extraction of near-synonyms from the Oxford English Dictionary. We then manually composed descriptive phrases that fully articulate the meaning and nuances of a particular motion. This approach creates a challenging set of prompts specifically designed to push the boundaries of the motion generation model's generalizability.

### C.3 NORMALIZATION FOR RADAR CHART VISUALIZATION

For the radar charts in Main Paper Fig. 1, we apply normalization to enable a fair and intuitive comparison of relative performance. Specifically, for each metric, we first scale all values by dividing them by the maximum value observed across models. For those metrics where smaller values indicate better performance, we then apply a positive transformation by taking the reciprocal of the scaled values. Finally, we normalize all metrics so that the best score among the models is mapped to 1.0 and the worst score is mapped to 0.2. The radar chart axes have a range of 0.0 to 1.0.

# D VIMOGEN-228K DATAILS

## D.1 VIMOGEN-228K VISUALIZATION

We provide some examples of our ViMoGen-228K in Fig. 7, including optical Mocap data, in-the-wild video data, and synthetic video data.

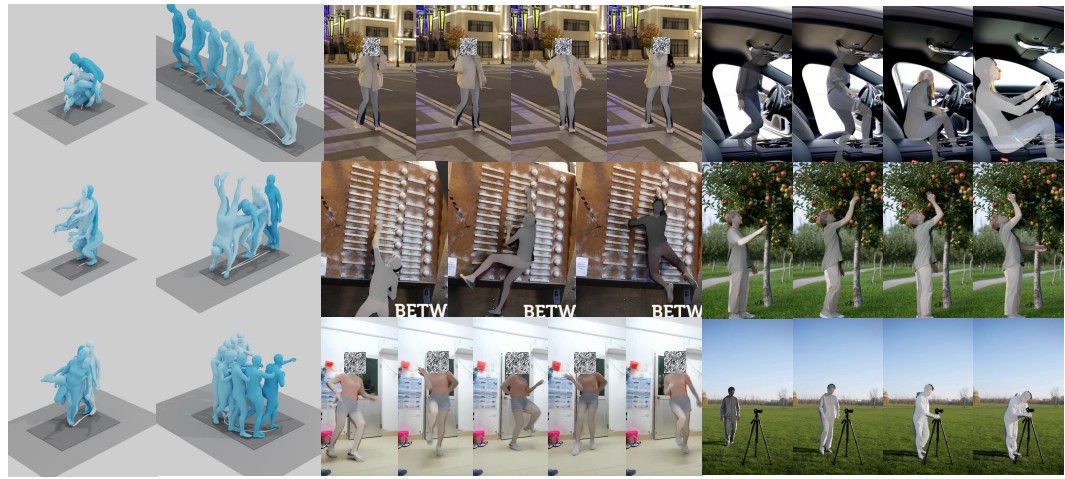

(a) Optical Mocap Data      (b) In-the-Wild Video Data      (c) Synthetic Video Data

Figure 7: Visualization of our ViMoGen-228K dataset. (a) High-fidelity Optical MoCap data. (b) Diverse In-the-Wild Video Data. (c) Precisely controlled Synthetic Video Data.

## D.2 VIMOGEN-228K COLLECTION, FILTERING, AND ANNOTATION

This section provides a detailed breakdown of the curation process for the ViMoGen-228K dataset, covering the collection, filtering, and annotation pipelines for each of our three primary data sources.

### D.2.1 OPTICAL MOCAP DATA

**Collection and Aggregation.** Our high-fidelity motion data is built upon a large-scale aggregation of 30 publicly available optical motion capture datasets. The initial pool of datasets includes: HumanML3D (Guo et al., 2022a), 100style (Mason et al., 2022), AIOZ-GDANCE (Le et al., 2023), AIST++ (Li et al., 2021), ARCTIC (Fan et al., 2023), BEAT (Liu et al., 2024), BEHAVE (Bhatnagar et al., 2022), Chairs (Jiang et al., 2023), CHI3D (Fieraru et al., 2025), ChoreoMaster (Chen et al., 2021), CIRCLE (Araújo et al., 2023), CMU (Tevet et al., 2023), EGO-BODY (Grauman et al., 2024), EMDB (Kaufmann et al., 2023), FineDance (Li et al., 2023c), FIT3D (Fieraru et al., 2021b), HOI-M3 (Zhang et al., 2024a), HumanSC3D (Fieraru et al., 2021a), IDEA400 (Lin et al., 2023), Interhuman (Liang et al., 2024b), InterX (Xu et al., 2024), KIT-ML (Plappert et al., 2016), LAFAN1 (Harvey et al., 2020), Mixamo, MoTORICA (Alexanderson et al., 2023), NeuralDome (Zhang et al., 2023b), OMOMO (Li et al., 2023a), PhantomDance (Li et al., 2022), RICH (Huang et al., 2022), and TRU-MANS (Jiang et al., 2024). All motion sequences were standardized to the SMPL-X (Pavlakos et al., 2019) format and resampled to a consistent 20 fps.

**Filtering and Selection.** To ensure high quality, we applied a rigorous multi-stage filtering pipeline. First, all long sequences were segmented into non-overlapping 5-second clips (100 frames). We then filtered out any clips shorter than 3 seconds and removed static T-pose frames. Subsequently, we used our MBench evaluation suite to programmatically assess motion quality, removing clips that exhibited significant jitter or lacked sufficient dynamics. Our dataset selection was an empirical, iterative process designed to maximize performance on downstream tasks. We began by establishing a baseline using the HumanML3D dataset, chosen for its high-quality, human-annotated text and its widespread use in the community. We then systematically evaluated the remaining 29 candidate datasets. For each candidate, we trained a text-to-motion model on a corpus combining the HumanML3D dataset with the candidate dataset. The performance of this model was then evaluated on our MBench

benchmark. This data-driven approach allowed us to quantify the contribution of each dataset. We observed that several datasets, particularly those with high-speed or stylistic dance motions and less semantic descriptive text, degraded overall model performance and text-motion consistency on MBench. This led us to exclude them from the final training set. Through this comprehensive curation process, we selected the 16 additional datasets that demonstrated a positive impact, resulting in a final corpus of 17 high-quality datasets (including the HumanML3D base) comprising 172k clips. The selected datasets are: **100style, ARCTIC, BEHAVE, Chairs, CIRCLE, EMDB, FIT3D, HumanML3D, HumanSC3D, IDEA400, InterX, KIT-ML, LAFAN1, Mixamo, OMOMO, RICH, and TRU-MANS**.

### D.2.2  IN-THE-WILD VIDEO DATA

**Collection.** To capture a diverse range of real-world motions, we sourced data from an internal mega-scale video database, which includes large public datasets such as Koala36M (Wang et al., 2024b), PANDA-36M (Chen et al., 2024), MIRA (Ju et al., 2024), and Ego-Exo (Grauman et al., 2024). This resulted in an initial pool of approximately 60 million video clips.

**Filtering.** The raw videos underwent a multi-stage filtering process. We first applied quality assessment filters to remove clips with low resolution, poor lighting, excessive motion blur, or heavy occlusion. This step yielded a more refined set of around 10 million clips suitable for further analysis. We then perform a second, more granular filtering pass. This process focused on selecting clips where at least 80% of the human skeleton is visible. This was supplemented by keyword searches for videos across categories like sports, daily activities, and cultural performances. While the current pipeline applies strict filtering to ensure the highest fidelity for model training and to balance between different data sources(Optical Mocap Data, In-the-Wild Video Data, Synthetic Video Data), we acknowledge the massive volume of discarded data. With the development of advanced MoCap algorithms (such as SAM 3D), we believe that the "imperfect" data can be used more efficiently in the future.

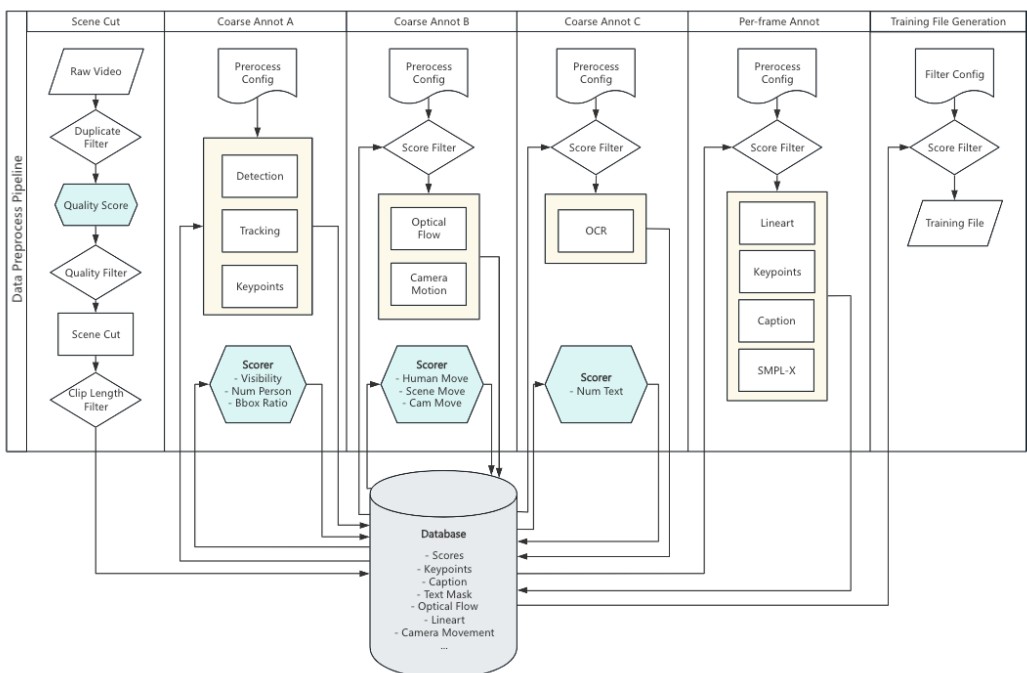

Figure 8: **Comprehensive Data Preprocess Pipeline.** The pipeline is organized into sequential modules, utilizing a central database to aggregate metadata like scores, keypoints, captions, and text masks.

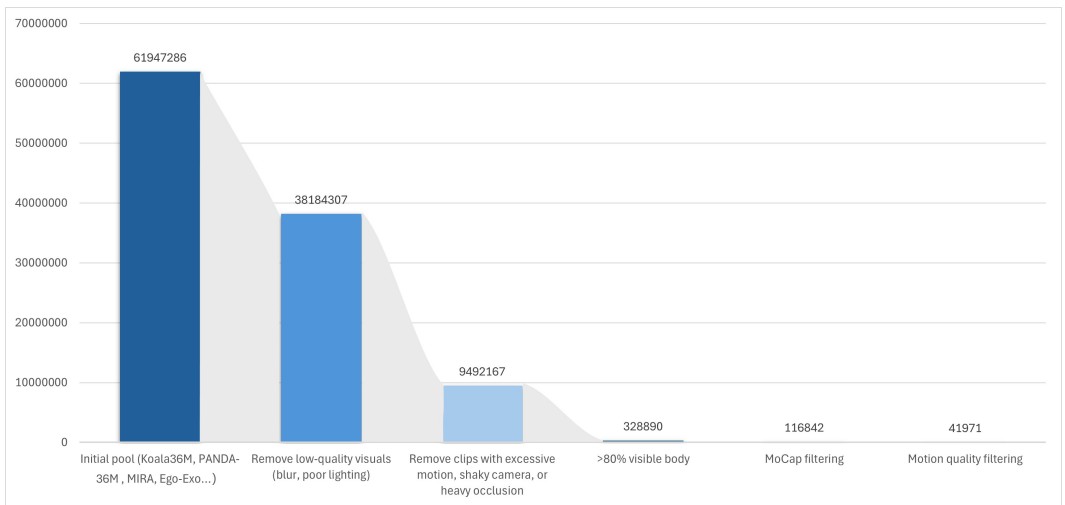

Figure 9: **Quantitative Results of the Filtering Process.**

### D.2.3 SYNTHETIC VIDEO DATA

**Prompt Construction.** To systematically expand the semantic coverage of our dataset, we leveraged the generative power of the state-of-the-art text-to-video model, Wan2.1 (Wan et al., 2025). We constructed a long-tail vocabulary by analyzing large-scale video caption datasets, including Open-HumanVid (Li et al., 2024) and LLaVA-Video-178K (Zhang et al., 2024c). From this vocabulary, we compiled a list of action verbs and descriptive nouns, which underwent semantic clustering and deduplication to produce over 20,000 unique prompts.

**Generation and Refinement.** These prompts were used with the Wan2.1 model to generate 81-frame video clips at 16 fps. We explicitly instructed the model to generate videos conducive to visual MoCap, such as those with a stable camera, a single full-body person, and a clean background. The generated videos and their extracted motions were passed through our standard filtering pipeline. We also interpolated the motion sequences from 16 fps to 20 fps linearly for alignment with common motion datasets. Crucially, the motions were further refined using our pre-trained ViMoGen Motion-to-Motion (M2M) branch to enhance fidelity, especially the world-space global translation that is usually estimated poorly by visual MoCap models. This process yielded a final set of 14,000 high-quality synthetic samples that fill critical semantic gaps in existing real-world datasets.

### D.2.4 DATA ANNOTATION

**Motion Annotation.** For all human video data (in-the-wild and synthetic data), we employed a two-stage pipeline. We first used the YOLOV8 tracking model (Jocher et al., 2023) to obtain human bounding boxes and then applied a state-of-the-art human mesh recovery model, CameraHMR (Patel & Black, 2025), to extract 3D motion in the SMPLX representation. Following DART (Zhao et al., 2025), we canonicalize the global orientation of each motion so that the initial frame faces the y+ axis. To mitigate common visual MoCap artifacts like jitter and foot-sliding, we applied post-processing smoothness algorithms based on temporal Gaussian smoothing.

**Text Annotation.** For MoCap datasets with existing high-quality annotations, such as HumanML3D, we used the provided text. For MoCap clips that lacked annotations or only had class labels, we generated descriptive text using a multi-modal LLM. We first rendered depth videos of the human mesh and provided these video frames to the Gemini 2.0 Flash model. For the in-the-wild videos, the original RGB clips were used as input. Our annotation process was guided by the detailed system prompt shown in Box 1, which was engineered to elicit structured and comprehensive motion descriptions. All LLM-generated annotations were heuristically filtered to remove invalid responses such as "I can't help with that" or "The speech content is: ...".

### D.2.5 System Prompt for Motion Description Annotation

We utilize the Gemini 2.0 Flash model to generate structured text annotations for our motion data. The model was provided with a sequence of rendered frames and the system prompt shown in Box 1. This prompt was carefully engineered to elicit detailed, multi-faceted descriptions covering holistic summaries, motion styles, and fine-grained temporal dynamics.

---

**Box 1: System Prompt for Gemini 2.0 Motion Description Annotation**

You are an expert in human motion analysis and biomechanics. Your task is to provide a detailed, structured annotation for a human motion sequence. You will be provided with a series of frames from a rendered video showing a 3D human motion. Analyze the sequence carefully and generate a description following the exact 7-part structure below. Be precise, objective, and use clear, descriptive language.

1. **Detailed description of whole human motion sequence:**
   Provide a comprehensive, narrative summary of the entire motion from start to finish. Describe the overall action, the flow of movement, and the purpose of the action if it is apparent.

2. **Styles, emotions of human motion sequence:**
   List 2-3 descriptive adjectives that capture the qualitative aspects of the motion. Examples include: *energetic, fluid, tense, relaxed, deliberate, clumsy, focused*.

3. **Description of the global trajectory and interaction with the environment:**
   Describe the overall path of the person's center of mass. Specify if the person is stationary (in-place) or moving through the environment. Note any interactions with the ground (e.g., foot contact, sliding) or implied objects.

4. **Key actions temporally:**
   Provide a concise, chronologically ordered list of the main actions or phases in the sequence. Use short verb phrases. For example: "Start standing, raise right leg, kick forward, return to standing."

5. **Action description every 2 frames:**
   Provide a fine-grained, frame-by-frame breakdown of the motion. For each 2-frame interval, describe the specific changes in body posture and limb positions.
   - Frame 1 to 2: [Description]
   - Frame 3 to 4: [Description]
   - ...and so on for the entire sequence.

6. **Action description every 4 frames:**
   Provide a slightly more coarse-grained breakdown of the motion. For each 4-frame interval, summarize the primary action or transition occurring within that window.
   - Frame 1 to 4: [Description]
   - Frame 5 to 8: [Description]
   - ...and so on for the entire sequence.

7. **Key motion status with corresponding essential body part movements:**
   Identify the most critical poses or phases of the motion (e.g., the peak of a jump, the moment of impact). For each key status, describe the configuration and movement of the essential body parts involved (e.g., torso, limbs, head).

---

Following the system prompt in Box 1, the model generates a structured annotation for each motion sequence. Box 2 provides a representative example for the motion of a person performing a drag curl. In our model training, we only use the information from Part 1.

---

**Box 2: Example of Generated Motion Description**

1. **Detailed description of whole human motion sequence:**
   The person starts in a semi-crouched position, with knees slightly bent and arms

---

close to the body. As the sequence progresses, the person gradually lowers their torso, moving from an upright position to a deeper squat, flexing at the hips and knees. The elbows remain bent, with the hands moving downward in a controlled motion, as if holding an object. The rhythm is steady and deliberate, emphasizing the smooth transition from a high to a low position, reflecting the controlled nature of the drag curl movement.

2. **Styles, emotions of human motion sequence:**
   Focused, deliberate.

3. **Description of the global trajectory and interaction with the environment:**
   The person stands in place, gradually lowering their torso and arms while maintaining balance on the spot.

4. **Key actions temporally:**
   Stand semi-crouched, lower torso and arms, deepen squat, maintain position.

5. **Action description every 2 frames:**
   - Frame 1 to 2: Bending slightly more at the knees, hands move downward.
   - Frame 3 to 4: Torso lowers further, arms start to extend slightly downward.
   - Frame 5 to 6: Knees continue to bend, torso moves forward and downward.
   - Frame 7 to 8: Torso reaches a lower position, hands nearing midpoint of descent.
   - Frame 9 to 10: Full squat position with arms fully extended downward.
   - Frame 11 to 12: Torso remains low, slight adjustment in arm position.
   - Frame 13 to 14: Maintaining position, minor adjustments to balance.

6. **Action description every 4 frames:**
   - Frame 1 to 4: A gradual increase in knee flexion, causing the torso to lower and extend arms downward.
   - Frame 5 to 8: Transition into deeper squat with arms moving in sync, maintaining alignment and balance.
   - Frame 9 to 12: Stabilization in the squatted position, ensuring balanced posture and extended arms.

7. **Key motion status with corresponding essential body part movements:**
   The person transitions smoothly from a standing position into a deep squat, characterized by a gradual lowering of the torso and extending arms downward. The upper body leans slightly forward to maintain balance, while the knees and hips are flexed deliberately, emphasizing stability and control in the motion sequence.

# E   VIMOGEN IMPLEMENTATION DETAILS

Our ViMoGen model builds upon the 1.3B-parameter Wan2.1 (Wang et al., 2025) text-to-video foundation model. Motion sequences are represented using the 276-dimensional SMPL-based vector from DartControl (Zhao et al., 2025).

**Training Configuration.** The model initializes with Wan2.1 weights and trains on 8 H800 GPUs using AdamW optimization (lr=0.0002, batch size=128) with FSDP for memory efficiency. Training ViMoGen on our MotionAtlas dataset requires 40,000 iterations (1.5 days), while ViMoGen-light completes in approximately one day.

**Adaptive Training Strategy.** To maximize the utility of heterogeneous data sources, we implement a data-aware training protocol. Synthetic data receives double weighting to compensate for potential quality variations, while visual mocap data supervision focuses only on local body poses to mitigate global motion artifacts. For branch selection during training, we adjust probabilities based on data quality: HumanML3D (Guo et al., 2022a) samples use 80% text-to-motion and 20% motion-to-motion generation, while other data uses balanced 40%/60% probabilities.

| Methods | R Precision↑ | | | FID↓ | MultiModal Dist↓ | MultiModality↑ |
|---|---|---|---|---|---|---|
| | Top 1 | Top 2 | Top 3 | | | |
| TM2T (Guo et al., 2022c) | $0.424^{\pm.003}$ | $0.618^{\pm.003}$ | $0.729^{\pm.002}$ | $1.501^{\pm.017}$ | $3.467^{\pm.011}$ | $2.424^{\pm.093}$ |
| T2M (Guo et al., 2022b) | $0.455^{\pm.003}$ | $0.636^{\pm.003}$ | $0.736^{\pm.002}$ | $1.087^{\pm.021}$ | $3.347^{\pm.008}$ | $2.219^{\pm.074}$ |
| MDM (Tevet et al., 2023) | $0.320^{\pm.005}$ | $0.498^{\pm.004}$ | $0.611^{\pm.007}$ | $0.544^{\pm.044}$ | $5.566^{\pm.027}$ | $\mathbf{2.799}^{\pm.072}$ |
| MotionDiffuse (Zhang et al., 2024b) | $0.491^{\pm.001}$ | $0.681^{\pm.001}$ | $0.782^{\pm.001}$ | $0.630^{\pm.001}$ | $3.113^{\pm.001}$ | $1.553^{\pm.042}$ |
| T2M-GPT (Zhang et al., 2023a) | $0.492^{\pm.003}$ | $0.679^{\pm.002}$ | $0.775^{\pm.002}$ | $0.141^{\pm.005}$ | $3.121^{\pm.009}$ | $1.831^{\pm.048}$ |
| MoMask (Guo et al., 2024) | $0.521^{\pm.002}$ | $0.713^{\pm.002}$ | $0.807^{\pm.002}$ | $\mathbf{0.045}^{\pm.002}$ | $2.958^{\pm.008}$ | $1.241^{\pm.040}$ |
| Motion-LCM (Dai et al., 2024) | $0.502^{\pm.003}$ | $0.698^{\pm.002}$ | $0.798^{\pm.002}$ | $0.304^{\pm.012}$ | $3.012^{\pm.007}$ | $2.259^{\pm.092}$ |
| MLD (Chen et al., 2023) | $0.481^{\pm.003}$ | $0.673^{\pm.003}$ | $0.772^{\pm.002}$ | $0.473^{\pm.013}$ | $3.196^{\pm.010}$ | $2.413^{\pm.079}$ |
| MLD + ViMoGen-light (Ours) | $\mathbf{0.542}^{\pm.003}$ | $\mathbf{0.733}^{\pm.002}$ | $\mathbf{0.825}^{\pm.002}$ | $0.114^{\pm.005}$ | $\mathbf{2.826}^{\pm.007}$ | $1.973^{\pm.074}$ |

Table 8: Quantitative evaluation on the HumanML3D test set. $\pm$ indicates a 95% confidence interval during 20 times repeating evaluations. **Bold** indicates the best result.

**Noise Simulation Parameters.** To simulate realistic visual MoCap errors for the M2M branch during training, we apply the following controlled noise parameters to the ground truth motion:

- **Gaussian Corruption:** Applied with a probability range of $[0.0, 0.4]$ and a scale range of $[0.05, 0.15]$.

- **Jitter Simulation:** We simulate temporal jitter by blending neighboring frames with a probability of $[0.03, 0.08]$ and a jitter strength of $0.3$.

- **Temporal Dropout:** Small temporal spans are masked with a rate of $0.02$ to mimic tracking loss.

Global translation is strictly masked for the M2M branch to eliminate domain gaps caused by trajectory estimation errors in monocular MoCap.

## F   TEXT-TO-MOTION EXPERIMENT ON HUMANML3D BENCHMARK

To further validate the effectiveness and generalizability of our proposed model architecture, we conduct a supplementary experiment on the widely-used HumanML3D benchmark (Guo et al., 2022a). This experiment isolates our core architectural contribution from our large-scale ViMoGen-228K dataset, demonstrating its strong performance in a traditional training setting. For this purpose, we integrate our network architecture into the established Motion Latent Diffusion (MLD) framework (Chen et al., 2023).

### F.1   EXPERIMENTAL SETTINGS

**Dataset.** We conduct this experiment on the **HumanML3D** (Guo et al., 2022a) dataset, a standard benchmark for text-to-motion generation. It contains 14,616 human motion sequences, paired with 44,970 descriptive text annotations. Following standard practice (Guo et al., 2022b; Dai et al., 2024), we utilize the same redundant motion representation, which includes root velocity and height, local joint positions, velocities, rotations relative to the root, and binary foot contact labels.

**Evaluation Metrics.** To ensure a fair and comprehensive comparison with prior work, we adopt the standard suite of evaluation metrics established by previous methods (Guo et al., 2022b; Chen et al., 2023). These metrics assess the generated motions from three key perspectives:

- **Motion Quality and Diversity:** We use the **Frechet Inception Distance (FID)** to measure the distributional similarity between generated and real motions. We also report **Diversity** and **MultiModality (MModality)** to evaluate the variety of motions generated from different texts and the same text, respectively.

- **Text-Motion Consistency:** We evaluate the semantic alignment between the generated motion and the input text using **R-Precision** (Top-1, Top-2, Top-3), which measures motion retrieval accuracy, and **Multimodal Distance (MM Dist)**, which calculates the average feature distance between text and motion pairs.

**Implementation Details.** Our implementation is built upon the reproduced version of MLD (Chen et al., 2023) from the official PyTorch codebase of MotionLCM (Dai et al., 2024). The core modification is the replacement of the original diffusion denoiser with our proposed text-to-motion network architecture. We leverage the pre-trained motion VAE from MotionLCM to operate within the same latent space, ensuring a fair comparison of the denoising network's capabilities.

To maintain consistency with the baseline, we retain most of the original hyperparameters from the code repository, including the choice of text encoder, the number of diffusion timesteps, and the Classifier-Free Guidance (CFG) scale used during inference. To optimize for our hardware and accelerate convergence, we tuned the training configuration by using a larger batch size and a correspondingly adjusted learning rate. The model was trained using the AdamW optimizer with a cosine learning rate scheduler for 36000 iterations.

## F.2 RESULTS AND ANALYSIS

We compare our method, ViMoGen-light, against a range of state-of-the-art models on the HumanML3D test set. The quantitative results are presented in Table 8. The experiment was repeated 20 times to report the mean and a 95% confidence interval, following standard evaluation protocol (Guo et al., 2022b; 2024).

As shown in the table, integrating our architecture into the MLD framework leads to a new state of the art in text-motion consistency. Our model, MLD + ViMoGen-light, surpasses all prior methods on every text-alignment metric, achieving the best R-Precision (Top-1, Top-2, Top-3) and the lowest Multimodal Distance. We attribute this significant leap in performance to our full-transformer denoiser, which enables a more nuanced and effective fusion of text and motion features compared to the original MLD architecture, thereby improving the interpretation of complex prompts. This superior text-motion alignment is further demonstrated in the qualitative examples presented in Fig. 10.

Furthermore, this substantial gain in semantic accuracy is achieved without sacrificing motion quality. Our model attains a highly competitive FID of 0.114, marking a dramatic improvement over the MLD baseline (0.473) and placing it on par with other top-performing methods like T2M-GPT. While maintaining a strong MultiModality score, these results validate that our architectural contributions are robust and independently effective. The experiment confirms that our approach is not only powerful when combined with our large-scale OmniMotion dataset but also serves as a potent, generalizable component that can advance existing frameworks on established benchmarks.

## G ADDITIONAL QUALITATIVE RESULTS

In this section, we provide additional qualitative results to complement the analysis presented in the main paper. Fig. 11 visualizes our adaptive branch selection mechanism in action, showcasing how ViMoGen intelligently switches between its M2M and T2M branches. Fig. 12 illustrates the impact of using descriptive *video-style* versus concise *motion-style* text for training and inference. Finally, Fig. 13 presents further side-by-side comparisons with state-of-the-art methods on MBench prompts, highlighting the superior semantic fidelity of our approach.

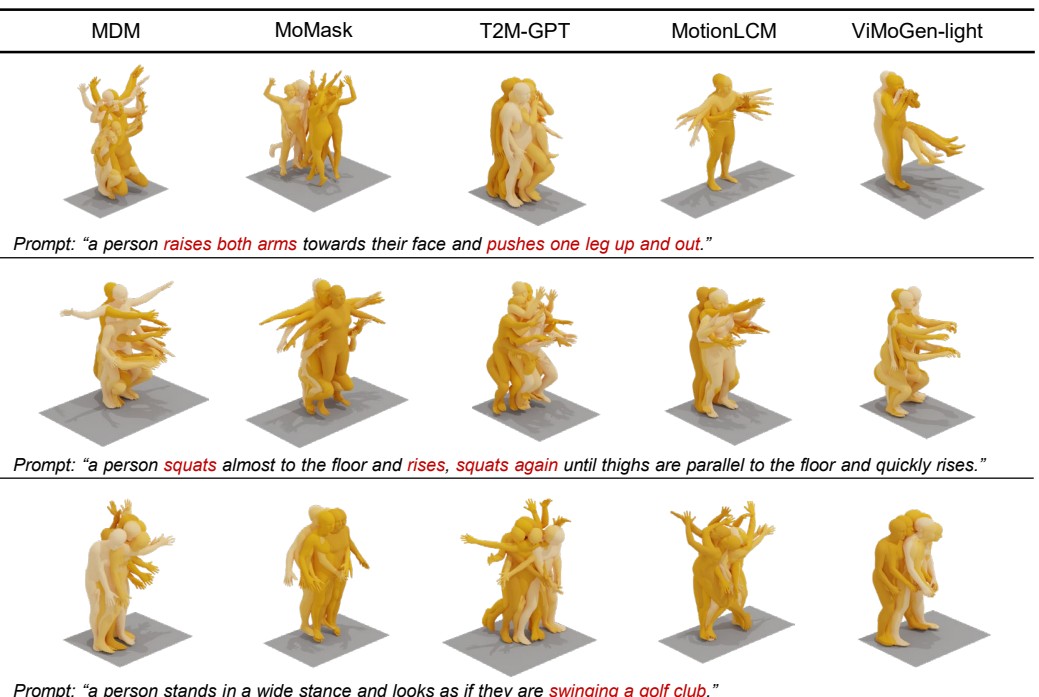

Figure 10: Qualitative comparison with state-of-the-art methods on the HumanML3D benchmark. For complex, multi-step prompts, our ViMoGen-light model generates motions that are more plausible and demonstrate superior text-motion alignment compared to prior works.

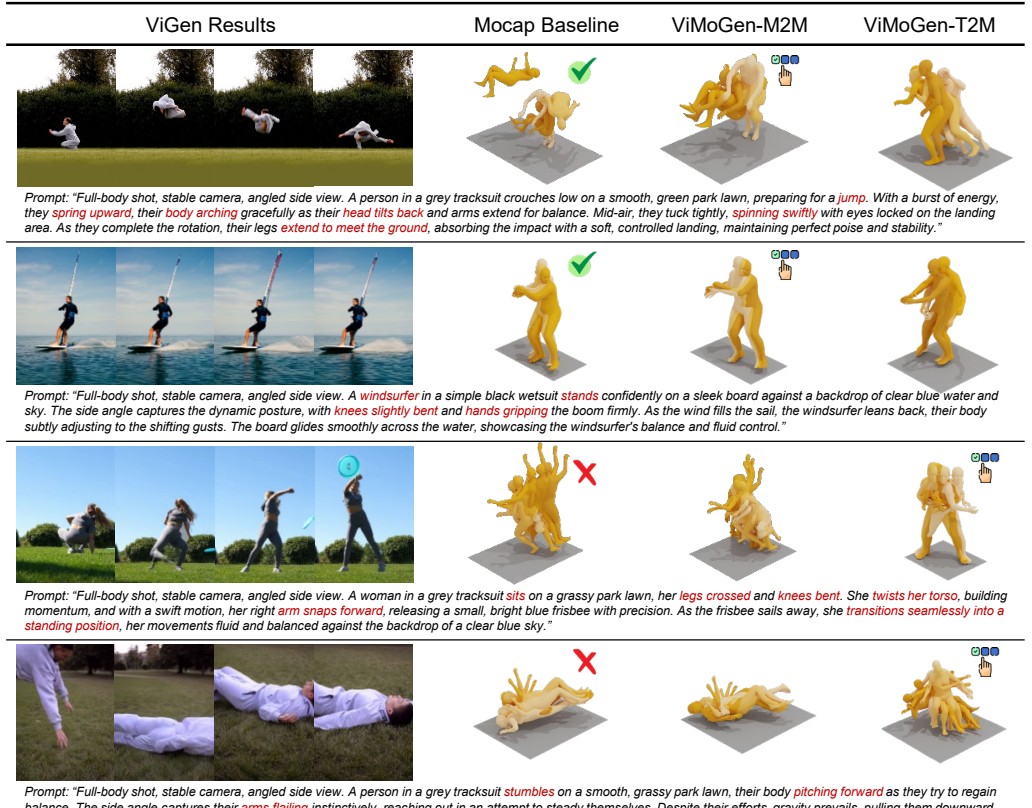

Figure 11: Qualitative examples of our adaptive branch selection mechanism. This figure showcases how ViMoGen intelligently chooses between its Motion-to-Motion (M2M) and Text-to-Motion (T2M) branches based on the quality of the initial motion extracted from generated videos (**Mocap Baseline**). **(Rows 1-2)** For prompts where the ViGen model produces a plausible motion sequence (*e.g.*, "backflip", "windsurfer"), the adaptive gate selects the **M2M branch**. This branch successfully refines the semantically correct but noisy Mocap Baseline, reducing jitter and improving physical realism. **(Rows 3-4)** For prompts involving sudden movements where the ViGen model fails and produces distorted or incomplete motions (*e.g.*, "twist and throw", "stumble and fall"), the Mocap Baseline is unreliable. Here, the adaptive gate correctly falls back to the more robust **T2M branch**, which generates a stable motion directly from the text prompt, ignoring the flawed video prior.

| Model Training Text | Testing Example1: swaggering into the room | | Testing Example2: chopping wood | |
| --- | --- | --- | --- | --- |
| | Video Style Prompt | Motion Style Prompt | Video Style Prompt | Motion Style Prompt |

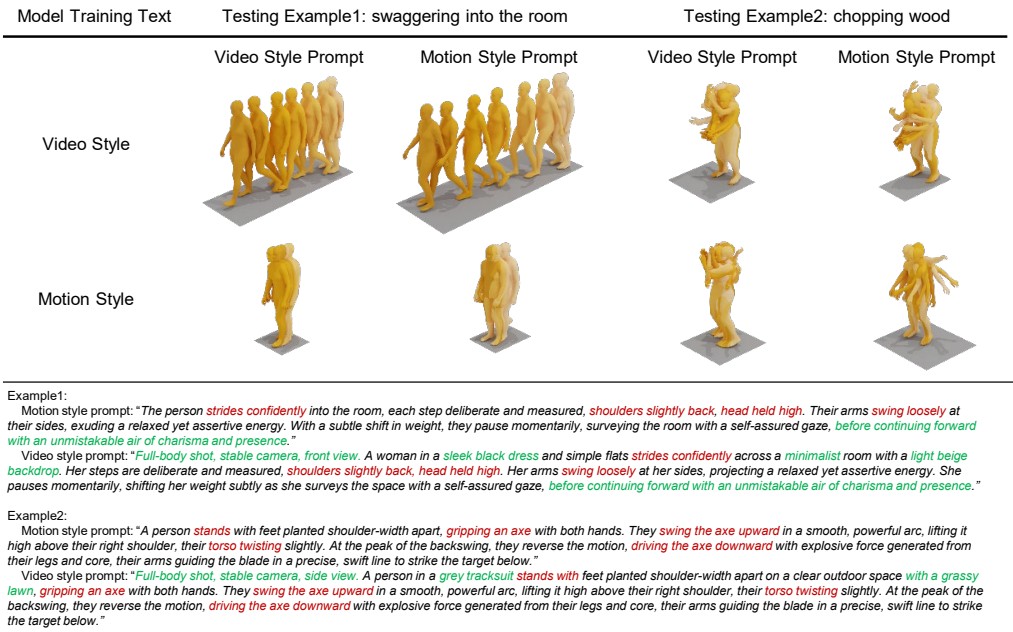

Example1:

Motion style prompt: "*The person strides confidently into the room, each step deliberate and measured, shoulders slightly back, head held high. Their arms swing loosely at their sides, exuding a relaxed yet assertive energy. With a subtle shift in weight, they pause momentarily, surveying the room with a self-assured gaze, before continuing forward with an unmistakable air of charisma and presence.*"

Video style prompt: "*Full-body shot, stable camera, front view. A woman in a sleek black dress and simple flats strides confidently across a minimalist room with a light beige backdrop. Her steps are deliberate and measured, shoulders slightly back, head held high. Her arms swing loosely at her sides, projecting a relaxed yet assertive energy. She pauses momentarily, shifting her weight subtly as she surveys the space with a self-assured gaze, before continuing forward with an unmistakable air of charisma and presence.*"

Example2:

Motion style prompt: "*A person stands with feet planted shoulder-width apart, gripping an axe with both hands. They swing the axe upward in a smooth, powerful arc, lifting it high above their right shoulder, their torso twisting slightly. At the peak of the backswing, they reverse the motion, driving the axe downward with explosive force generated from their legs and core, their arms guiding the blade in a precise, swift line to strike the target below.*"

Video style prompt: "*Full-body shot, stable camera, side view. A person in a grey tracksuit stands with feet planted shoulder-width apart on a clear outdoor space with a grassy lawn, gripping an axe with both hands. They swing the axe upward in a smooth, powerful arc, lifting it high above their right shoulder, their torso twisting slightly. At the peak of the backswing, they reverse the motion, driving the axe downward with explosive force generated from their legs and core, their arms guiding the blade in a precise, swift line to strike the target below.*"

Figure 12: Qualitative examples illustrating the impact of different text prompt styles used during training and inference. The rows represent the style of text used to train the model (*video-style* vs. *motion-style*), while the columns show the prompt style used for testing. We display results for two prompts: "swaggering into the room" and "chopping wood." The visualizations demonstrate that the model trained on descriptive *video-style* text (top row) is more robust and generalizes better, producing high-quality motions for both concise *motion-style* and rich *video-style* prompts at test time. Full prompt examples are provided below the images for clarity.

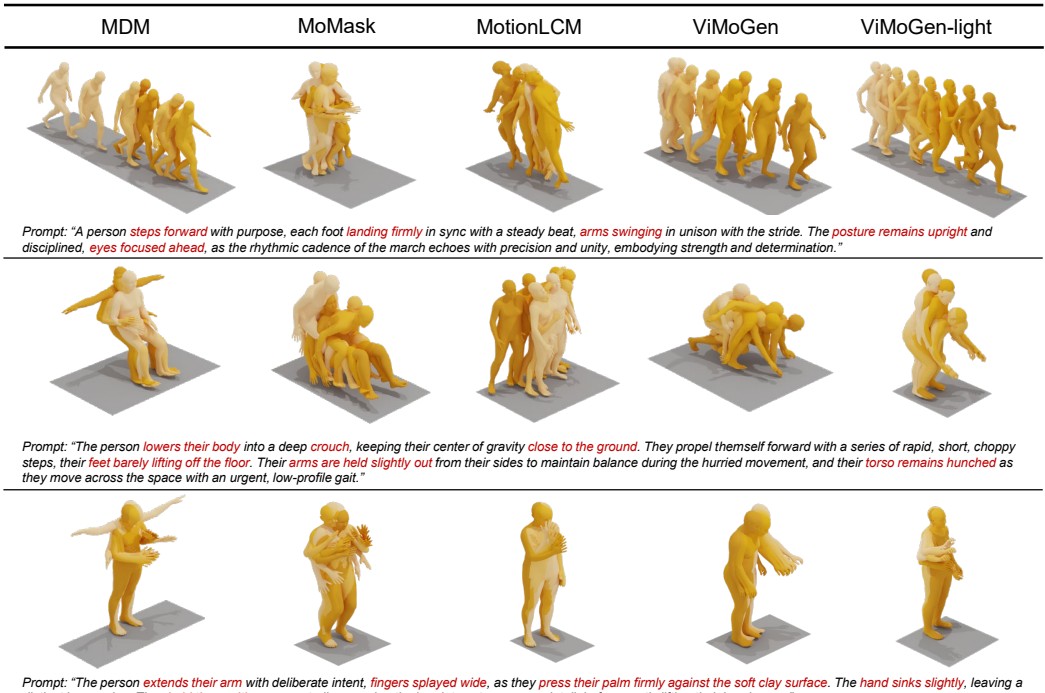

Figure 13: Additional qualitative comparisons with state-of-the-art methods on MBench prompts. Both ViMoGen and ViMoGen-light consistently generate motions that more faithfully adhere to the detailed text descriptions, showcasing their superior semantic understanding and generation quality compared to prior works.

