# OpenReview forum: "The Quest for Generalizable Motion Generation: Data, Model, and Evaluation"
_ICLR.cc/2026/Conference — ICLR 2026 Poster_

### Official Review · Reviewer_b56g · 2025-10-30

**Soundness:** 3
**Presentation:** 3
**Contribution:** 3
**Rating:** 8
**Confidence:** 3

**Summary:**

This paper proposes a comprehensive framework of data, models, and evaluation to address the generalization capability of 3D human motion generation (MoGen). It leverages prior knowledge from ViGen.

For data, the authors built ViMoGen-228K, a large-scale dataset with 228,000 motion clips. It innovatively fuses high-fidelity optical motion capture (MoCap) data, diverse data from web videos, and long-tail data synthesized by Video Generation (ViGen) models. This wide variety of actions helps improve the generalization of MoGen.

For the model, the authors proposed ViMoGen, a flow-matching-based diffusion Transformer. It uses a novel gated dual-branch (T2M and M2M) architecture to adaptively unify the quality priors from MoCap data and the generalization priors from ViGen models. A distilled, efficient version, ViMoGen-light, is also provided.

For evaluation, the authors designed MBench, a new hierarchical benchmark for the comprehensive and fine-grained assessment of motion quality, text-motion consistency, and especially generalization capability.

**Strengths:**

Originality:

1. To expand semantic coverage, the dataset leverages a Video Generation (ViGen) model to synthesize long-tail motion data, which is then integrated with traditional MoCap data.

2. The paper introduces MBench, a hierarchical benchmark meticulously designed for the fine-grained evaluation of motion generalization capabilities, featuring a curated open-world vocabulary.

Quality:

1. The paper presents a complete and systematic solution—spanning data collection, model architecture, and a novel evaluation benchmark—demonstrating a thorough and solid investigation.

2. The method's effectiveness is rigorously validated through extensive experiments, including comprehensive comparisons against state-of-the-art (SOTA) methods and detailed ablation studies. Furthermore, the MBench metrics are corroborated by a large-scale human preference study, ensuring their alignment with human judgment.

Clarity:
1. The paper is well-structured and clearly delineates its three core contributions: the dataset (ViMoGen-228K), the model (ViMoGen), and the benchmark (MBench).

2. The manuscript includes high-quality figures that effectively aid comprehension. For instance, Figure 1 clearly illustrates the overall framework and comparative radar charts; Figure 2 intuitively presents the model's dual-branch gated architecture; and Figure 3 vividly details the evaluation dimensions of MBench.

Significance:

1. This work identifies and addresses a critical bottleneck in the field of 3D human motion generation: poor generalization capability. It proposes an effective and integrated solution to tackle this fundamental challenge.

2. The authors commit to the public release of their code, the ViMoGen-228K dataset, and the MBench benchmark. These artifacts will serve as a valuable public resource, poised to stimulate further research and development in this area.

**Weaknesses:**

1. ViMoGen-228K leverages a mixture of three data sources: high-fidelity MoCap, alongside in-the-wild and synthetic videos. The filtering process for the in-the-wild data yields an extremely low retention rate (a reduction from 60M to 40k clips). This raises the question of whether in-the-wild video data, in itself, is inherently unsuitable for human motion extraction. Compounding this, based on Table 5, the "Visual Mocap Data" component does not appear to yield significant improvements.

2. Regarding the generalization capabilities discussed (relative to ViGen), the 228k (370h) dataset used for MoGen is still considerably smaller than ViGen's pre-training corpus. How do the authors think MoGen's generalization capabilities compared with ViGen's generalization?

3. The primary quantitative results (Table 2) are reported on MBench, a benchmark concurrently introduced by the authors. While MBench appears to be well-designed, this presents a potential risk: namely, that the new data and the proposed model may have "overfit" to the specific evaluation criteria of this new benchmark.

**Questions:**

1. Regarding the data filtering pipeline, neither the main paper nor Appendix D.2 provides a detailed procedure. Key details are missing, such as the specific quality assessment filters employed and the retention ratios (or absolute clip counts) at each filtering stage. This lack of transparency makes the drastic data reduction (from 10M to 40k clips) , raises concerns.

2. In Appendix D.2.3 (Synthetic Data), the authors state they "compiled a list of action verbs and descriptive nouns."
It is recommended that the authors provide this complete list in the appendix and, crucially, conduct a comparative analysis against the vocabulary used in MBench (to ensure the generation vocabulary does not overly align with the benchmark).
Additionally, clarification is required on how potentially ambiguous verbs (e.g., "have") were programmatically handled or disambiguated during text prompt generation.

3. ViMoGen-light: Table 2 reveals a substancial performance gap between ViMoGen-light and the full ViMoGen. The primary source of this discrepancy is not clearly analyzed.
Furthermore, the distillation (referred to as "distill") used to create ViMoGen-light is critically underspecified. What specific distillation methodology was employed (e.g., DMD2, or )?
What performance benefits  provide by ViMoGen-light (e.g., computation cost)?

**Details Of Ethics Concerns:**

The authors' plan to publicly release the code, the ViMoGen-228K dataset, and the MBench benchmark is a commendable contribution to the research community. However, this release necessitates a careful ethical and legal review, particularly concerning the dataset's licensing.

---

> ### Author Response · Authors · 2025-11-21
> **Response to Reviewer b56g**
>
> Dear Reviewer b56g:
>
> Thank you for your supportive review and recognition of our method's strengths. We hope the following revisions and clarifications address your remaining concerns.
>
> **Q-1: Visual MoCap data contribution is limited.**
>
> **A-1:** We acknowledge that the immediate quantitative impact of visual MoCap data is currently constrained in our experiments.
>
> - **Challenge:** Existing visual mocap models still struggle with complex scenarios (e.g., occlusion, multi-person interaction), necessitating aggressive filtering that reduces the effective data volume and semantics coverage.
> - **Future Potential:** However, we regard this pipeline as a "proof-of-concept" for future scaling. As visual MoCap accuracy improves, our framework can immediately ingest larger-scale video data without architectural changes.
> - **Community Value:** Furthermore, we believe the curated Text-Motion-Video triplet dataset itself is a significant contribution, providing a valuable resource for future multimodal motion research.
>
> **Q-2: Motion data is still far smaller than video data.**
>
> **A-2:** We recognize this domain gap, where video generation (ViGen) models generalize far better than motion generation (MoGen) models due to training data scale. We bridge this gap using two strategies to transfer ViGen capabilities to MoGen:
>
> 1. **Data Synthesis:** We utilize SOTA ViGen models to generate diverse videos from rich text prompts, then extract motion to create a synthetic training set. This expands the semantic coverage of our motion model.
> 2. **Dual-Branch Fusion:** Our architecture explicitly fuses video priors into the motion generation process.
>
> As demonstrated in Tab.3 and Tab. 5, both strategies significantly improve the model's generalization capabilities compared to training on motion data alone.
>
> **Q-3: Risk of new data and model overfitting to MBench.**
>
> **A-3:** We have taken specific measures to ensure our model remains robust and generalizable, rather than overfitting to MBench:
>
> 1. **Objective Metrics:** MBench relies primarily on physics-based, objective metrics (e.g., Foot Sliding, Jitter, Body Penetration) rather than distribution-based metrics (like FID relative to a specific set). These metrics evaluate physical plausibility, which is dataset-agnostic.
> 2. **Prompt Diversity:** Our test prompts are carefully designed to be semantically distinct from the training set to prevent memorization.
>
> **Q-4: Details of visual MoCap data filtering.**
>
> **A-4:** Thank you for the reminder! We have added more details about the visual MoCap data filtering pipeline in the revised manuscript (see Appendix D.2.2).
>
> Fig. 8 **(Supplementary Material 3)** shows the comprehensive data preprocess pipeline, detailing the specific quality assessment filters and logical flow used to curate the dataset.
>
> Fig. 9 **(Supplementary Material 4)** provides the absolute clip counts and retention breakdown at each filtering stage.
> While the current pipeline applies strict filtering to ensure the highest fidelity for model training and to balance between different data sources (Optical Mocap Data, In-the-Wild Video Data, Synthetic Video Data), we acknowledge the massive volume of discarded data. With the development of advanced MoCap algorithms (such as SAM 3D), we believe that the "imperfect" data can be used more efficiently in the future.
>
> **Q-5: Questions about synthetic data.**
>
> **A-5:** Our synthetic dataset is built from a vocabulary of approximately 20,000 action verbs and descriptive nouns. While space constraints prevent listing them all in the paper, the full list will be released with the code. Our construction pipeline ensures quality by:
>
> 1. **De-duplication:** We perform retrieval against the MBench vocabulary and rewrite or remove terms that are too semantically similar to the test set.
> 2. **Semantic Filtering:** We use an LLM to verify semantic clarity, filtering out abstract words that do not correspond to clear, executable physical actions.
>
> **Q-6: Details about distillation.**
>
> **A-6:** We apologize for the confusion. We clarify that we employ Knowledge Distillation, not Step Distillation like DMD.
>
> - **Goal:** The objective is to create a lightweight model (*ViMoGen-light*) that does not require the computationally expensive video generation branch during inference.
> - **Method:** We treat the full *ViMoGen* model (which uses T2V + M2M branches) as the Teacher. It generates high-quality pseudo-ground-truth motions from synthetic text prompts. The *ViMoGen-light* (Student) model—which only contains the T2M branch—is then trained on this high-quality synthetic data using standard flow matching objectives. This allows the lightweight model to internalize the priors from the video-guided teacher.

---

> > ### Comment · Reviewer_b56g · 2025-11-25
> >
> > I thank the authors for their response. Regarding my primary concern about the "Details of visual MoCap data filtering," I found the answer in Figures 8 and 9. I have no further questions. I suggest open-sourcing the project as soon as possible, as it would be a valuable contribution to the research community.

---

> > > ### Author Response · Authors · 2025-11-25
> > > **Thank you for your feedback**
> > >
> > > Thank you for your positive assessment and valuable suggestion. We are currently organizing the dataset and will proceed with open-sourcing data, code, and benchmark as soon as it is ready.

---

### Official Review · Reviewer_ixmY · 2025-10-31

**Soundness:** 2
**Presentation:** 2
**Contribution:** 2
**Rating:** 4
**Confidence:** 5

**Summary:**

This paper proposes a comprehensive framework to enhance generalization in 3D human motion generation by transferring knowledge from video generation.
The contributions are threefold.
1) The ViMoGen-228K dataset is introduced, integrating 228,000 high-quality motion samples from optical MoCap, web videos, and synthetic ViGen data to expand semantic diversity.
2) The ViMoGen model is presented, a diffusion transformer through flow matching that unifies MoCap and ViGen priors via gated multimodal conditioning, alongside a distilled variant, ViMoGen-light, for efficient inference.
3) MBench, a hierarchical benchmark, is developed for fine-grained evaluation across motion quality, prompt fidelity, and generalization. Experiments demonstrate superior performance over existing methods.

**Strengths:**

The strengths of this paper lie in the following aspects:
1) The construction of the ViMoGen-228K dataset demonstrates is encouraging, combining high-fidelity optical MoCap data with semantically diverse motions from in-the-wild and synthetically generated videos.
The multi-stage filtering pipeline encourages a balance between motion quality and semantic coverage.

2) The design of the ViMoGen model's gated, dual-branch diffusion transformer is interesting as well. The adaptive selection mechanism between the text-to-motion and motion-to-motion branches dynamically balances high-quality MoCap priors with the broad semantic knowledge from ViGen models.

3) The paper is very easy to understand and easy to follow. A straightforward presentation of both the data-set and methodology.

4) As for the experiments, I notice the empirical gains on MBench: higher motion-condition consistency (0.53) and generalizability (0.68) than MDM, T2M-GPT, MotionLCM, and MoMask; reduced jitter (0.0108) and foot sliding (0.0064) with analyzed trade-offs in dynamic degree.

**Weaknesses:**

My major concern is that
1) this paper lack of key comparison with methods:
(i) MotionCraft (Bian et al., 2025) which shows state-of-the-art performance on text-to-motion on the HumanML3D subset of Motion-X dataset (Lin et al., 2023)
(ii) FineMoGen(Zhang et al., 2023),
(iii) MotionDiffuse both of which are also representative recent approaches and show promising results.
As well, the comparison is only conducted on the proposed MBench benchmark. How about the proposed approach on the widely adopted benchmarks? e.g. text-to-motion on the HumanML3D subset of Motion-X dataset; speech-based motion generation on the BEAT2 dataset; music-based motion generation on the FineDance.

2) The topic and focus of this paper is too broad, while motion generation, has many aspects: speech-based, text-to-motion, music-based. I recommend the authors to narrow down the focus and scope and propose the advantages and difference the previous proposed approaches.

References:

[1] Yuxuan Bian, Ailing Zeng, Xuan Ju, Xian Liu, Zhaoyang Zhang, Wei Liu, and Qiang Xu. Motioncraft: Crafting whole-body motion with plug-and-play multimodal controls. In Proceedings of the AAAI Conference on Artificial Intelligence, volume 39, pp. 1880–1888, 2025.

[2] Mingyuan Zhang, Huirong Li, Zhongang Cai, Jiawei Ren, Lei Yang, and Ziwei Liu. Finemogen: Fine-grained spatio-temporal motion generation and editing. Advances in Neural Information Processing Systems, 36: 13981–13992, 2023c.

[3] Mingyuan Zhang, Zhongang Cai, Liang Pan, Fangzhou Hong, Xinying Guo, Lei Yang, and Ziwei Liu. Motiondiffuse: Text-driven human motion generation with diffusion model. IEEE transactions on pattern analysis and machine intelligence, 46(6):4115–4128, 2024b.

Some other technical problems:
1) There may exist potential data contamination and selection bias: (i) synthetic video motions are “refined” with a pre-trained ViMoGen M2M branch used later for training, creating circularity; (ii) optical MoCap subset selection is optimized using MBench, risking overfitting the training data to the proposed evaluation.
2) How about the gating mechanism robustness? branch selection depends on VLM-based alignment to video-derived motion; training substitutes real video references with noise-perturbed GT motions, introducing a distribution gap between training and inference.
3) How about the text annotation validity? large portions of text labels are produced by an MLLM from rendered depth/RGB frames with heuristic filtering only; no inter-annotator agreement or systematic quality audit is provided, risking noisy supervision for text–motion alignment.
4) The dataset provenance and ethical are recommended to clarify: the internal video pool and synthetic data prompts are derived from large external caption corpora; consent, licensing of source material, and redistribution rights of derivative 3D motions are insufficiently detailed.

**Questions:**

Please see the weakness section.

---

> ### Author Response · Authors · 2025-11-21
> **Response to Reviewer ixmY**
>
> Dear Reviewer ixmY:
>
> We sincerely appreciate your constructive feedback, which has significantly improved the quality of our manuscript. We believe the following revisions and clarifications effectively address your concerns.
>
> **Q-1: Comparison with related work on MBench and conventional benchmarks.**
>
> **A-1:** Thank you for this suggestion. We have expanded our evaluation to include MotionCraft, FineMoGen, and MotionDiffuse on both MBench and conventional benchmarks.
>
> - **MBench Performance:** As shown in the table below, our method demonstrates a significant lead in *Motion Condition Consistency* and *Motion Generalizability* compared to the baselines, while maintaining comparable performance on quality-related metrics.
>
> |Method|Motion-Condition Consistency↑|Motion Generalizability↑|Jitter Degree↓|Dynamic Degree↑|Foot Floating↓|Foot Sliding↓|Body Penetration↓|Pose Quality↓|
> |-|-|-|-|-|-|-|-|-|
> | FineMogen | 0.37 | 0.42 | 0.0118 | 0.0386 | 0.281 | 0.0091 | 1.177 | 2.279 |
> | MotionDiffuse | 0.44 | 0.42 | 0.0111 | 0.0289 | 0.126 | 0.0063 | 1.352 | 2.214 |
> | MotionCraft | 0.42 | 0.45 | 0.0132 | 0.042 | 0.402 | 0.009 | 1.151 | 2.118 |
> | ViMoGen (ours) | 0.53 | 0.68 | 0.0108 | 0.0251 | 0.204 | 0.0064 | 1.775 | 2.382 |
>
> - **Conventional Benchmarks (HumanML3D Testset):** We report results for our method (*ViMoGen-light*) against the suggested baselines in the table below. Our approach achieves competitive results on standard metrics. *Note:* As our method focuses specifically on text-to-motion generation, we did not conduct experiments on speech-based or music-based generation benchmarks. We have clarified this scope in the revised manuscript.
>
> |Method|R Precision Top 1↑|R Precision Top 2↑|R Precision Top 3↑|FID↓|MultiModal Dist↓|MultiModality↑|Diversity↑|
> |-|-|-|-|-|-|-|-|
> | FineMogen | 0.504 | 0.690 | 0.784 | 0.151 | 2.998 | 2.696 | 9.263 |
> | MotionDiffuse | 0.491 | 0.681 | 0.782 | 0.630 | 3.113 | 1.553 | 9.410 |
> | MotionCraft | 0.501 | 0.697 | 0.796 | 0.173 | 3.025 | - | 9.543 |
> | ViMogen-light (ours) | 0.542 | 0.733 | 0.825 | 0.114 | 2.826 | 1.973 | 9.453 |
>
>
>
> **Q-2: Considering narrowing the scope.**
>
> **A-2:** This is a valid point. We have revised the Abstract and Introduction to explicitly state that our work focuses on text-to-motion generation. We clarify that speech-based and music-based generation are outside the scope of this paper. If deemed necessary by the committee, we are open to updating the title to specify "Text-to-Motion Generation" to reflect this focus more accurately.
>
> **Q-3: Data contamination and selection bias.**
>
> **A-3:** We address the concerns regarding data distribution as follows:
>
> - **M2M branch diversity**: The M2M branch is trained on a massive corpus of MoCap data collected from diverse sources. This scale ensures the model does not overfit to any single domain and maintains high diversity in motion priors.
> - **No bias from filtering**: As mentioned in Section D.2.1, we utilize MBench to filter optical MoCap data. Crucially, this filtering relies on objective, physics-based metrics (e.g., detecting body penetration or severe jitter) rather than data-driven or model-based likelihoods. Consequently, this process removes low-quality artifacts without introducing semantic selection bias to the data distribution.

---

> ### Author Response · Authors · 2025-11-21
> **Continued Response to Reviewer ixmY (Q4-Q6)**
>
> **Q-4: Robustness of the gating mechanism.**
>
> **A-4:** We validate the robustness of our VLM-based gating mechanism against noise:
>
> - **Experimental Setup:** Following denoising protocols in *HuMoR* and *DNO*, we add Gaussian noise ($\sim$4cm) to the reference motion of 450 samples. We then visualize these noisy motions (at 5 fps) and feed them to the VLM (Gemini-2.0-Flash) to see if the gating decision changed.
> - **Stability:** The VLM decision remains consistent for 80.4% of samples. This stability arises because the VLM focuses on high-level semantic alignment in the video rather than high-frequency motion jitter.
> - **Impact on Performance:** We measure the final metrics when using this "noisy gating" (inputting noisy video to VLM, but keeping the original reference for the M2M branch to isolate the gating effect). As shown below, the performance drop is minimal, confirming our gating is robust to input quality variations.
>
> |Setting|Motion-Condition Consistency↑|Motion Generalizability↑|Jitter Degree↓|Dynamic Degree↑|Foot Floating↓|Foot Sliding↓|Body Penetration↓|Pose Quality↓|
> |-|-|-|-|-|-|-|-|-|
> | Gating with additional noise | 0.51 | 0.65 | 0.0112 | 0.0262 | 0.212 | 0.0065 | 1.611 | 2.318 |
> | Original ViMoGen metrics | 0.53 | 0.68 | 0.0108 | 0.0251 | 0.204 | 0.0064 | 1.775 | 2.382 |
>
>
> **Q-5: Text annotation validity.**
>
> **A-5:** We ensure high-quality text annotations through a two-step process:
>
> 1. **Hierarchical Annotation:** We employ a coarse-to-fine strategy where the VLM first labels the overall action category and then provides a detailed decomposition of body movements.
> 2. **Manual filtering**: We manually review samples from each MoCap source. For highly dynamic or abstract subsets where accurate text description is inherently difficult (e.g., complex modern dance), we exclude them from the T2M training set. These samples are used exclusively for the M2M branch to learn motion priors without confusing the text-motion alignment in the T2M branch.
>
> **Q-6: Clarify provenance and ethics.**
>
> **A-6:** Thank you for this important reminder regarding data ethics and provenance. We rigorously adhere to the licensing terms of all source datasets and have designed our release strategy to ensure full compliance:
>
> - **Video Data:** To respect the copyright and privacy policies of platforms (e.g., YouTube), we will not distribute raw video files. Instead, we will release a metadata file containing video URLs and timestamps, requiring users to query the content directly from the source.
> - **MoCap Data:** We strictly respect the intellectual property and licenses of the original optical MoCap datasets. Upon release, users will be required to verify their adherence to the original data licenses (and sign necessary usage agreements where applicable) before they are granted access to download our processed motion data.

---

### Official Review · Reviewer_4ZNa · 2025-10-31

**Soundness:** 3
**Presentation:** 3
**Contribution:** 3
**Rating:** 6
**Confidence:** 3

**Summary:**

This paper addresses the limited generalization of current 3D human motion generation models and proposes a unified framework that transfers knowledge from video generation models. They introduce ViMoGen-228K, a large-scale dataset combining high-fidelity MoCap data, in-the-wild video motions, and synthetic ViGen-generated samples to enhance semantic diversity. Besides, the authors present a flow-matching diffusion-based model with adaptive gating to fuse MoCap and ViGen priors. Finally, an evaluation benchmark MBench, is proposed for comprehensive evaluation.

**Strengths:**

* The paper addresses the generalization limitation in 3D human motion generation by offering a comprehensive solution across data, models, and evaluation benchmarks.
* The paper is well-written and easy to follow, with well-organized structure.
* The authors introduce ViMoGen-228K, a large-scale and diverse motion dataset, and MBench, a fine-grained evaluation benchmark. This provides training and evaluation solutions for the field of motion generation, which is beneficial for advancing technological development in this domain.

**Weaknesses:**

* Despite the authors claims strong generalization, the paper does not thoroughly examine where the model fails (e.g., on highly dynamic, multi-person scenarios). Analysis on this part is expected.
* While the model is excels at the generalization, there appears to be a trade-off, as it does not outperform all baselines on certain motion quality metrics like dynamic degree. Is there any solution to alleviate this problem?
* MBench relies heavily on VLM-based automatic scoring and curated prompts. I'm curious about the metric‘s robustness and sensitivities. Has the author analyzed these factors?

**Questions:**

Please refer to weaknesses section.

---

> ### Author Response · Authors · 2025-11-21
> **Response to Reviewer 4ZNa**
>
> Dear Reviewer 4ZNa:
>
> We thank the reviewer for the positive rating. Please refer to the following response, which we hope addresses your questions.
>
>
> **Q-1: Discussion of failure cases.**
>
> **A-1:** We appreciate this valuable suggestion. We have added a dedicated discussion on failure cases in the revised manuscript. Our method currently faces two primary limitations:
>
> * **Single-Person Constraint:** Our current architecture is designed for single-person motion generation and does not support multi-person interactions.
> * **Complex High-Dynamic Motions:** The model occasionally struggles with highly dynamic, in-the-wild motions (e.g., a "gymnastic full twist"). This limitation stems from our reliance on video generation models for initialization; when the base video generator fails to render these complex dynamics accurately, our M2M branch may not fully correct the errors.
>
> **Q-2: Trade-off between generalization and motion quality.**
>
> **A-2:** You raise a very important point. While our model achieves superior generalization, it does not surpass all baselines on specific motion quality metrics. This trade-off is primarily driven by two factors:
>
> * **Visual MoCap Quality:** Current visual MoCap datasets inherently contain noise and artifacts (such as foot sliding), which impacts the precision of the trained model.
> * **Video Generation Dynamics:** Video generation models sometimes produce motions with limited dynamic range, which constrains the "Dynamic Degree" of our outputs.
>
> To address this in future work, we propose: 1) Incorporating advanced MoCap algorithms with explicit foot contact constraints to improve data quality, and 2) adjusting the training strategy to force the model to extrapolate high-quality dynamics even from low-dynamic video initializations.
>
> **Q-3: Robustness of MBench metrics.**
>
> **A-3:** We carefully design MBench with robustness in mind:
>
> 1. **Objective metrics dominance:** Only two metrics (motion generalizability and motion-condition consistency) use VLM; most others rely on objective metrics (e.g., foot floating, jitter degree).
> 2. **VLM robustness strategies:** As mentioned in lines 366-370, we use a multiple-choice format with chain-of-thought prompting. The VLM first describes observed motion, then selects from ten candidates: one ground-truth and nine distractors. These distractors are carefully designed rather than randomly chosen. They are formed using cosine similarity thresholds from three percentile ranges (below 5th, 47-52nd, above 95th), ensuring semantic variability and consistent evaluation.
> 3. **Experimental validation:**
>    * These metrics show high alignment with human preferences, which are relatively reliable (revised Fig. 6, **Supplementary Material 2**).
>    * We test robustness by varying rendering conditions and candidate words. Results show metrics are not significantly affected by rendering conditions or candidate word changes, demonstrating robustness (see table below showing Motion Condition Consistency scores across different rendering textures: 0.473, 0.53, 0.53, with minimal variation).
>
> | Setting | Score |
> |-|-|
> | Render texture 1 (**Supplementary Material 6**) | 0.473 |
> | Render texture 2 (**Supplementary Material 6**) | 0.53 |
> | Render texture 3 (official setting) | 0.53 |
> | One ground-truth + 8 distractors | 0.50 |
> | One ground-truth + 9 distractors (official setting) | 0.53 |

---

> > ### Comment · Reviewer_4ZNa · 2025-11-26
> >
> > Thank the author for your efforts in responding. Most of my concerns have been addressed. Besides, I suggest including a more comprehensive discussion and visualization regarding failure cases. I decide to keep my initial score.

---

> > > ### Author Response · Authors · 2025-11-27
> > > **Thank you for your feedback**
> > >
> > > We sincerely thank the reviewer for the constructive feedback and valuable suggestions. We will add a dedicated discussion about failure cases with visual examples in the revision.

---

### Official Review · Reviewer_gdGg · 2025-10-31

**Soundness:** 2
**Presentation:** 2
**Contribution:** 3
**Rating:** 4
**Confidence:** 5

**Summary:**

The authors aim at bridging video generation and 3D human motion generation by proposing a new large-scale dataset ViMoGen-228k, a novel diffusion transformer ViMoGen for unifying the video priors and mocap priors, and a new benchmark MBench for evaluating motion generalizability, motion-condition consistency, and motion quality.

**Strengths:**

1. Contribution a large-scale motion datasets with 228,000 motion sequences, including high fidelity MoCap data and automatically annotated motion datas from web videos and synthesized videos, which is great for pushing forward the field of motion generation.

1. Great efforts on trying to provide a robust and quantifiable evaluation benchmark for motion generalizability and motion-condition consistency. The proposed based on gated multimodal conditioning is proven to be effective under the proposed benchmark.

**Weaknesses:**

1. Poor presentation for qualitative results.  Take Fig.4 for example, for several samples, the SMPL frame shots are highly overlapped and the resolution are too low for the readers to compare the visual qualities, e.g. doing jumping jacks, martial arts, climbing a ladder. Similar issues can be observed in the qualitative results figures throughout the paper. Also, the axe or reference for temporal direction should be mentioned alongside.

1. No demo videos are provided. Given the limitations of static frame shot figures, it’s crucial to provide convincing video demos of the generated samples and dataset samples to justify the qualitative comparison. However, the authors haven’t provided any video demos in supplementary materials.

1.  In the proposed gating diffusion block, the T2Mbranch seems to be totally parallel with the M2Mbranch, performing totally different tasks with only one shared self-attention layer.  And the adaptive branch selection at inference relies heavily on alignment scores from external VLM to determine which branch to go with. It’s then questionable whether the two branch can really benefit from each other in learning the underlying distribution, or the performance is mainly gained from the external VLM assessment and selection for the task to proceed with.

1. The training pipeline is not clear. The curriculum approach mentioned in line 204 - 207 is worth elaborated:
    - “we simulate video motion references by perturbing ground-truth motions with controlled noise.” — What is the controlled noise mentioned here? And it’s very confusing to me how perturbing the ground-truth motions can simulate the video motion references.
    - For training with text-motion pairs of high quality MoCap data,  would there still be a z_video for M2Mbranch? If so, how to ensure the alignment of the z_video and ground-truth high quality motion? The video generated from the same text may contain totally different motion from the ground-truth motion.

1. The details of inference phase is not clear either:
    - How to assess if the alignment scores from VLM is high or low? Through a scalar threshold?
    - What are the differences and advantages of using the M2Mbranch, other than extracting the MoCap motion directly from the generated videos and performing motion refinement with a SOTA refinement method? Any empirical results on this?

1. The benchmark metrics haven’t been fully evaluated whether it aligns with human perception or not. Although Appendix C.1.2 presents the alignment results for the temporal motion quality, the human preference study for the abstractive metrics (motion generalizability and motion-condition consistency) based on VLM are not provided.

1. The frame-wise quality seems to replace the old distribution-based metrics (e.g. FID) with another “new” distribution-based metrics, except that the definition for the underlying distribution is different. Why would it be better to set up this way? Moreover, any empirical results of why this whole set of motion quality metrics is more reliable and robust than before?

1. The baselines and ablations in Table 3 are not very convincing. Practically, the M2Mbranch is playing the role of a motion refinement module in this context, so the compared baseline should additionally includes text-to-video +  a SOTA motion refinement, to clearly evaluate the power of M2Mbranch.

1. To fairly ablate the impact of external VLM assessment during inference, there should also be a specific ablation for not using the gated diffusion block, and instead only use the external VLM alignment scores to select from a SOTA T2M model and a SOTA text-to-video + motion refinement model, and other two ablations with replacing the one of two models with T2M or M2M respectively.

1. Please pay attention to the typos, e.g. line 110, line 363.

**Questions:**

Please refer to the weaknesses section for details. If the identified issues and questions are properly addressed, I would consider raising my score.

---

> ### Author Response · Authors · 2025-11-21
> **Response to Reviewer gdGg**
>
> Dear Reviewer gdGg:
>
> We sincerely appreciate your constructive suggestions which helped us revise our paper to strengthen it. We hope the following revisions and clarifications can address the reviewer's remaining concerns:
>
> **Q-1:** Poor presentation of qualitative results.
>
> **A-1:** Thanks for your kind advice! We have updated Fig.4 **(Supplementary Material 1)** in the revision with higher resolution images, reduced frame overlap, and clear temporal axis annotations.
>
> **Q-2:** Lack of a demo video.
>
> **A-2:** We have prepared comprehensive demo videos in the supplementary materials, showing the quality of the generated samples and dataset samples.
>
> **Q-3:** Whether T2M and M2M branches benefit each other?
>
> **A-3:** Thank you for highlighting this crucial aspect of our method. We clarify the mutual benefits through architectural design, data synergy, and empirical results:
>
> 1. **Shared architecture**: The T2M and M2M branches share approximately 66% of the DiT parameters, including all Self-Attention and Feed-Forward Network (FFN) layers. Only the Cross-Attention layers are task-specific.
> 2. **Mutual Benefits via Data Synergy:** Joint training allows the model to learn more robust motion priors by merging data from both tasks:
>     - **Rich Motion Priors:** Optical MoCap data often lacks text annotations but contains high-quality motion dynamics. While this data is primarily used for M2M training, the shared parameters allow the T2M branch to internalize these motion priors, enhancing the plausibility of text-generated motions.
>     - **Balanced Training Strategy:** To implement this, we adjust the task probabilities based on dataset characteristics. For HumanML3D (manually annotated), we set the probability to 80% T2M and 20% M2M. For other datasets (VLM-annotated), we use a 40% T2M / 60% M2M split. Crucially, because the weights are shared, the model "sees" all motion patterns uniformly, regardless of the specific task branch active at a given step.
> 3. **Empirical Validation:** To verify this, we conduct the baseline experiment suggested in Q9 (comparing our joint approach against a selection-based baseline using VLM alignment scores). We also performed ablations, replacing one branch with a SOTA baseline (*MotionLCM*) or a motion refiner (*DNO*[1]). As shown in the table below, our joint training approach consistently outperforms the baselines in motion consistency and quality metrics.
>
> |T2M branch|M2M branch|Motion-Condition Consistency↑|Motion Generalizability↑|Jitter Degree↓|Dynamic Degree↑|Foot Floating↓|Foot Sliding↓|Body Penetration↓|Pose Quality↓|
> |-|-|-|-|-|-|-|-|-|-|
> |Ours|Ours|0.53|0.68|0.0108|0.0251|0.204|0.0064|1.775|2.382|
> |Ours|DNO|0.50|0.65|0.0138|0.0320|0.255|0.0075|1.725|2.453|
> |MotionLCM|Ours|0.49|0.66|0.0179|0.0374|0.217|0.0136|1.870|2.483|
> |MotionLCM|DNO|0.49|0.63|0.0205|0.0432|0.297|0.0217|1.984|2.567|
>
> **Q-4:** Clarification of the training pipeline.
>
> **A-4:**
>
> 1. We clarify the noise simulation and branch handling mechanisms as follows:
>
> - **Controlled Noise Strategy:** To bridge the gap between clean MoCap data and noisy video inference, we apply a compound corruption procedure during training that mimics vision-based estimation errors. Instead of simple Gaussian noise, we employ:
>
>     - **Random Corruption:** Gaussian noise is added with a probability of [0.0, 0.4] and a scale of [0.05, 0.15].
>     - **Jitter Simulation:** To mimic short-term jitter, we replace frames with a blend of neighboring frames (probability [0.03, 0.08], jitter strength 0.3).
>     - **Temporal Dropout:** We randomly mask small temporal spans (rate 0.02) to simulate tracking loss.
> - **Domain Gap Handling:** Since visual MoCap typically produces accurate local poses but unreliable global translation, we mask the global translation during both training and inference. The model relies solely on local motion, effectively eliminating the global trajectory domain gap.
> 2. **$Z_{video}$ Handling:** We activate strictly one branch per sample:
>   - **T2M Active:** No video latent $z_{video}$ is input to the M2M branch.
>   - **M2M Active:** We simulate video motion references by perturbing ground-truth motions using the controlled noise strategy described above (rather than generating synthetic videos). This ensures $z_{video}$ remains semantically aligned with the ground truth while being robust to noise.
>
> References:
>
> [1] Karunratanakul, Korrawe, et al. "Optimizing diffusion noise can serve as universal motion priors." Proceedings of the IEEE/CVF Conference on Computer Vision and Pattern Recognition. 2024.

---

> > ### Comment · Reviewer_gdGg · 2025-11-23
> >
> > Thanks to the authors for the efforts of adding the evaluation experiments and providing detailed clarifications to address my earlier questions. I now have a more clearer understanding of the methodology and the benchmark design. However, I still have concerns regarding the perceptual generation quality, given the very few generated motion clips presented in the demo video. In the current Supplementary Material 5_Demo_Video, it appears that only three generated motion clips are provided, while none are corresponding to the strong qualitative samples shown in the paper figures, and all three are based on very short text prompts (e.g. pull ups, karate) which seems not challenging for most of the existing works.  Could you provide more qualitative generated motion clips in video form, including both short (e.g. Figure 4) and long (e.g. Figure 10) text prompts? If possible, it would also be better to include the motion clips in video form for your comparison with previous methods in Figure 4 and Figure 10.

---

> ### Author Response · Authors · 2025-11-21
> **Continued Response to Reviewer gdGg (Q5-Q7)**
>
> **Q-5:** Clarification of the inference pipeline.
>
> **A-5:**
>
> 1. **Alignment Assessment:**
>     - **Setup:** We use Gemini-2.0-Flash to process visualized videos of the rendered human mesh (sampled at 5 fps).
>     - **Protocol:** The VLM is prompted to perform a binary check: *Does the rendered motion align with the text description?*
>         - **If Yes:** The reference motion is deemed usable; we refine it using the M2M branch.
>         - **If No:** The reference is discarded; we generate motion from scratch using the T2M branch.
>     - **Robustness:** While a multi-VLM voting ensemble could increase precision, we intentionally utilize this simple discrimination method to demonstrate the robustness of our framework—even with a basic selector, our unified model yields high-quality results.
> 2. **M2M branch advantages**: As detailed in the following Table, our M2M branch significantly outperforms standard refinement baselines. The quantitative comparison below shows that our method effectively reduces jitter and foot artifacts while maintaining high consistency.
>
>
> |Setting|Motion Condition Consistency↑|Motion Generalizability↑|Jitter Degree↓|Dynamic Degree↑|Foot Floating↓|Foot Sliding↓|Body Penetration↓|Pose Quality↓|
> |-|-|-|-|-|-|-|-|-|
> | Reference motion | 0.51 | 0.58 | 0.0193 | 0.039 | 0.363 | 0.0161 | 1.831 | 2.536 |
> | DNO refiner | 0.45 | 0.58 | 0.0212 | 0.0448 | 0.473 | 0.0151 | 2.169 | 2.688 |
> | M2M (ours) | 0.51 | 0.59 | 0.0145 | 0.0309 | 0.273 | 0.0113 | 1.825 | 2.545 |
>
> **Q-6:** Human preference study for the abstractive metrics.
>
> **A-6:** We originally did not include Motion Generalizability and Motion-Condition Consistency in our human-alignment analysis because these two metrics rely on VLM-based evaluations, which inherently resemble human perceptual judgments. While evaluation dimensions such as foot floating or foot sliding are physics-based and continuous measures, it was unclear how closely they would correlate with human preferences.
>
> In the revised Fig. 6 **(Supplementary Material 2)**, we include human preference comparisons for all MBench dimensions. The scatter plots show strong positive correlations between human-preference win ratios and MBench evaluation win ratios, with Spearman ρ values reaching up to 1.0. Notably, the Foot Floating dimension has a lower correlation. This is because human generally have limited sensitivity to subtle foot-floating artifacts, causing their preference scores to cluster around 0.5. In contrast, the physics-based metric in our benchmark can precisely distinguish these differences.
>
> As mentioned in Sec. 4.1, our Motion Generalizability and Motion-Condition Consistency are evaluated differently from the Motion Quality pillar. These two dimensions are measured through multiple-choice tasks, where the VLM first describes the motion, then selects the most matching text description from 10 carefully selected text options. The selection is correct, the sample scores; otherwise, it doesn't. This results in binary correctness rather than continuous [0,1] scores. Consequently, when computing MBench win ratios, two methods that both receive either 0 or 1 for a sample result in a tie, which reduces the variation in their win-ratio distribution.
>
> **Q-7:** Comparison with previous motion quality metrics.
>
> **A-7:**  Thanks for your question! In fact, among the seven dimensions of Motion Quality evaluation, six of them are physics-based metrics, and we only use one distribution-based metric (NRDF) for evaluating Pose Quality. NRDF is not the same as previous metrics like FID; it measures the quality of each individual pose instead of the entire datasets, enabling fine-grained evaluation of motion quality frame-by-frame. In the revised Fig. 6 (Supplementary Material 2), the scatter plots show strong positive correlations between human-preference win ratios and MBench evaluation win ratios for motion quality metrics. This indicates that our physics-based metrics capture the same qualitative differences that human observers rely on when judging motion realism. We also conduct a human preference ranking study on 100 randomly selected HumanML3D test samples regarding motion naturalness and quality.  The results are summarized below:
>
> |Method|Human Ranking|FID Ranking|Jitter Degree (Ours)|Foot Floating (Ours)|Foot Sliding (Ours)|
> |-|-|-|-|-|-|
> |MDM|1|4|1|1|1|
> |MotionLCM|3|3|3|4|3|
> |T2M-GPT|4|2|4|3|4|
> |MoMask|2|1|2|2|2|
>
>  Results show our metrics align with human ranking, while FID ranking diverges significantly from human perception.

---

> ### Author Response · Authors · 2025-11-21
> **Continued Response to Reviewer gdGg (Q8-Q10)**
>
> **Q-8**: Add a baseline in Table 3.
>
> **A-8**: Thank you for this suggestion. We have added a baseline consisting of a three-stage pipeline: Text-to-Video (WanVideo) $\rightarrow$ Motion Capture $\rightarrow$ Motion Refinement (DNO).
>
> - **Implementation:** For DNO, we utilized its official optimization framework and parameters. Unlike the original DNO paper, which tests on synthetic Gaussian noise, we input the raw reference motion extracted from the video without adding extra artificial degradation.
> - **Results Analysis:** As shown in the table above (Q-5), the DNO refiner actually leads to performance degradation (higher jitter and lower consistency) compared to the raw reference motion, whereas our M2M method improves quality.
> - **Reason for DNO’s Failure:** This highlights a critical domain gap. Visual MoCap errors are structurally different from the Gaussian noise typically assumed by refiners like DNO. Specifically, MoCap models (like CameraHMR) often estimate accurate 2D reprojections but produce highly unstable global translations, particularly in videos with partial body visibility (e.g., upper-body shots). Methods like DNO enforce keypoint constraints that inadvertently force the motion to adhere to this erroneous global trajectory, resulting in severe artifacts.
> - **Our Advantage:** In contrast, our M2M branch is explicitly designed to handle visual MoCap noise (as detailed in A-4). By masking global translation and training on a specific noise profile that mimics MoCap errors, our model successfully refines the motion while preserving semantic integrity.
>
> **Q-9:** Comparison with another baseline.
>
> **A-9:** Please refer to A-3 for the detailed comparison results. We implemented the suggested baseline using external VLM alignment scores to select between a SOTA T2M model and a "Text-to-Video + Motion Refinement" pipeline. We also compared this against ablations where the selection pool included our specific T2M or M2M branches. The results confirm that our unified joint-training approach achieves higher consistency and motion quality than simply selecting between disparate pre-trained models.
>
> **Q-10:** Typos.
>
> **A-10:** We appreciate your attention to detail. We have corrected the typos at line 110 and line 363 in the revised manuscript.

---

> ### Author Response · Authors · 2025-11-24
>
> Thank you for your constructive feedback! We have updated the supplementary demo video to answer your questions about perceptual generation quality. The updated video now includes more qualitative comparisons with SOTA methods across diverse prompt types, including motion prompts from MBench (Figure 4), traditional benchmark prompts from HumanML3D (Figure 10), and three in-the-wild cases. We want to clarify that for MBench and in-the-wild prompts, we use long motion descriptions as input but only display keywords in the captions for simplification, as noted in Figure 4's caption. For visualizing comparison methods, we convert their results from skeleton-based HumanML3D-263-dim representation to SMPL parameters using the IK algorithm from the official T2M-GPT repository. We hope these updates address your concerns, and please let us know if you have any further questions.

---

> > ### Comment · Reviewer_gdGg · 2025-11-25
> >
> > Much thanks to the authors for the updated qualitative results and the detailed description, which are presented to be quite convincing and impressive. My concerns have been well addressed by the authors' replies, and I would like to increase my score upon these.

---

> > > ### Author Response · Authors · 2025-11-25
> > > **Thank you for your feedback**
> > >
> > > Thank you for your positive reassessment, We truly appreciate the time and effort you invested in reviewing our submission.

---

### Author Response · Authors · 2025-11-22
**Response to All Reviewers and Area Chairs**

We sincerely thank all reviewers and area chairs for their time and valuable comments. All source code, pre-trained models, dataset, and benchmark will be made publicly available for further research.

Reviewers recognize our work's strengths: the comprehensive framework spanning data, model, and evaluation (Reviewer gdGg, Reviewer 4ZNa, Reviewer b56g), the large-scale ViMoGen-228K dataset with 228,000 motion clips (Reviewer gdGg, Reviewer 4ZNa, Reviewer ixmY, Reviewer b56g), the novel gated dual-branch architecture for unifying video and motion priors (Reviewer 4ZNa, Reviewer ixmY, Reviewer b56g), fine-grained MBench evaluation benchmark (Reviewer gdGg, Reviewer 4ZNa), and significant improvements in generalization capability (Reviewer gdGg, Reviewer 4ZNa, Reviewer ixmY).

In response to the concerns raised, we have provided comprehensive demo videos in supplementary materials showcasing dataset samples and generated motions. We have carefully responded to all questions raised by the reviewers with detailed explanations and additional experiments. The revised manuscript includes improved figure quality, expanded experimental comparisons, and corrections to all typographical errors. All feedback has been integrated as detailed in our individual responses to each reviewer.

---

### Author Response · Authors · 2025-11-30
**Summary of Rebuttal and Discussion Phase**

Dear Area Chair and Reviewers,

We sincerely thank all reviewers and the area chair for your time and valuable feedback throughout the review process. As the discussion phase concludes, we would like to provide a summary of our efforts and the outcomes.

**1. Recognized Strengths**

Reviewers acknowledge several key strengths of our work across our main contributions.

- The **comprehensive framework** spanning data, model, and evaluation is recognized by Reviewers gdGg, 4ZNa, and b56g.
- All reviewers appreciate the **large-scale ViMoGen-228K dataset** with 228,000 motion clips from diverse sources.
- The **novel gated dual-branch architecture for unifying video and motion priors** is highlighted by Reviewers 4ZNa, ixmY, and b56g.
- Additionally, the **fine-grained MBench evaluation benchmark** is recognized by Reviewers gdGg and 4ZNa, while the **significant improvements in generalization capability** are acknowledged by Reviewers gdGg, 4ZNa, and ixmY.

**2. Major Concerns and Our Revisions**

Reviewers raise several important concerns that we address comprehensively.

- Regarding the lack of **qualitative visualization**, we provide a comprehensive **demo video** showcasing both dataset samples and generated motions, and **improve figure quality** throughout the manuscript.
- To address questions about whether the **T2M and M2M branches mutually benefit** each other, we conduct detailed ablation studies with quantitative results demonstrating that **our joint training approach** consistently **outperforms baseline alternatives** including simple model selection and single-branch hybrids.
- Concerns about limited **baseline comparisons** are resolved by expanding our evaluation to include **MotionCraft, FineMoGen, and MotionDiffuse** on both **MBench** and the standard **HumanML3D** benchmark.
- For concerns regarding **MBench's reliability and human alignment**, we perform detailed correlation analysis across all evaluation dimensions, demonstrating **strong positive correlations** between automated metrics and human preferences.
- Regarding the **scope** of our work, we have explicitly clarified in the revised Abstract and Introduction that our primary focus is on **text-to-motion generation**, ensuring the contribution aligns precisely with the presented evaluations.
- Additionally, we clarify the **training pipeline details** and add comprehensive documentation of our **data filtering process** with quantitative retention statistics at each stage.

**3. Reviewer Responses**

The response from reviewers has been highly encouraging.

- **Reviewer gdGg** acknowledges that their **concerns are well addressed** after a three-round discussion and **raises their score from 4 to 6**, stating "**My concerns have been well addressed by the authors' replies, and I would like to increase my score.**"
- **Reviewer 4ZNa** confirms that "**Most of my concerns have been addressed**" and maintains their **positive score of 6**.
- **Reviewer b56g** expresses satisfaction with our responses and maintains their **strong positive score of 8**.
- Regarding **Reviewer ixmY**, although **no response** has been received yet, we have comprehensively addressed all raised concerns including **comparisons with suggested baselines** and **clarifications on scope**. If deemed necessary by the committee, we are willing to update the title to specify "Text-to-Motion Generation" to better reflect our focused contribution.

**4. Commitment to Further Improvement**

All source **code, pre-trained models, dataset, and benchmark** will be made publicly available to facilitate future research. If there are any remaining concerns or questions from the reviewers or area chair, we are fully committed to addressing them promptly and thoroughly.

Thank you again for your time and valuable feedback throughout this process!

Sincerely,

The Authors

---

### Meta-Review · Area_Chair_6nsP · 2026-01-02

**Summary:**

Reviewers agree the paper is ambitious and potenitally impactful in framing "generalizable motion generation" as a unified effort across data (ViMogen-228k), modeling (dual-branch gated diffusion transformer + ViMoGen-light), and evaluation (MBench). The main points of contention were not about whether the direction is meaningful, but about **whether the claims are sufficiently supported** and **whether the proposed benchmark/model design is rigorous and transparent**.

Key concerns driving the lower scores included: (i) initially insufficient qualitative evidence (limited video demos / unclear figures), (ii) unclear training/inference details of the gating/branching pipeline and how “video priors” are simulated during training, (iii) whether the two branches genuinely provide mutual benefit vs. relying on external VLM-based selection, (iv) adequacy of baseline comparisons on both the new benchmark and conventional benchmarks, and (v) reliability/robustness of the VLM-based evaluation components in MBench.

**Reviewer Concerns:**

### Concerns addressed by the rebuttal
* **Missing/weak qualitative evidence**: The authors added more demo videos and improved qualitative visualization quality, addressing the concern that static figures were insufficient.
* **Baseline coverage and evaluation on standard benchmarks**: In response to requests for comparisons to MotionCraft / FineMoGen / MotionDiffuse and results beyond MBench, the authors report expanded comparisons on both MBench and HumanML3D (via ViMoGen-light), and narrow down the paper’s focus to text-to-motion rather than speech/music.
* **MBench robustness / human alignment**: They report correlation analysis between automated metrics and human preferences and also discuss robustness checks (e.g., varying rendering/candidate choices for the VLM-based component).
* **Failure cases and trade-offs**: They add an explicit discussion of limitations/failure modes (e.g., struggles with highly dynamic motions; single-person constraint; dependence on upstream video generation quality), and acknowledge trade-offs in certain motion quality metrics.
* **Provenance/licensing clarity**: They state an intended release approach that avoids distributing raw videos and enforces license compliance for MoCap sources.

### Outstanding concerns
* **Reliance on VLM-based gating and VLM-based scoring components**: Even with robustness checks, there remains a structural concern that key parts of the pipeline (branch selection and parts of evaluation) depend on specific VLM behavior and prompting choices, which may affect reproducibility and stability across settings.
* **Data contamination / circularity concerns (raised by the most critical reviewer)**: The rebuttal addresses parts of the selection-bias point, but deeper questions about possible circularity in synthetic data refinement and benchmark-aligned filtering may not be fully settled within the current evidence.

**Reviewer Scores:**

Reviewer gdGg actively participated in the rebuttal discussion which led to a promised score raising (4->6). Reviewer b56g (rating 8) and Reviewer 4ZNa (rating 6) would likely stay the original score, since they explicitly state most concerns were addressed. Reviewer ixmY would likely increase the score to 6. Several major points were directly addressed by the rebuttal (expanded comparions + clarified scope).

---

### Decision · Program_Chairs · 2026-01-26

Accept (Poster)